# Cancer-cell-secreted miR-204-5p induces leptin signalling pathway in white adipose tissue to promote cancer-associated cachexia

Yong Hu[1,2,5], Liu Liu[1,5], Yong Chen [1], Xiaohui Zhang[1], Haifeng Zhou[1], Sheng Hu[1], Xu Li[1], Meixin Li [1], Juanjuan Li[3], Siyuan Cheng[2], Yong Liu [1,4], Yancheng Xu [2] ✉ & Wei Yan [1] ✉

Cancer-associated cachexia is a multi-organ weight loss syndrome, especially with a wasting disorder of adipose tissue and skeletal muscle. Small extracellular vesicles (sEVs) serve as emerging messengers to connect primary tumour and metabolic organs to exert systemic regulation. However, whether and how tumour-derived sEVs regulate white adipose tissue (WAT) browning and fat loss is poorly defined. Here, we report breast cancer cell-secreted exosomal miR-204-5p induces hypoxia-inducible factor 1A (HIF1A) in WAT by targeting von Hippel-Lindau (*VHL*) gene. Elevated HIF1A protein induces the leptin signalling pathway and thereby enhances lipolysis in WAT. Additionally, exogenous VHL expression blocks the effect of exosomal miR-204-5p on WAT browning. Reduced plasma phosphatidyl ethanolamine level is detected in mice lack of cancer-derived miR-204-5p secretion in vivo. Collectively, our study reveals circulating miR-204-5p induces hypoxia-mediated leptin signalling pathway to promote lipolysis and WAT browning, shedding light on both preventive screenings and early intervention for cancer-associated cachexia.

In breast cancer (BC), the most commonly diagnosed female cancer worldwide[1], the incidence of cachexia is reportedly 19–40%[2,3]. Cancer-associated cachexia is characterised as systemic inflammation and multi-organ wasting disorder that accompanies weight loss, especially with mass loss in adipose tissue and skeletal muscle[4,5]. Cachexia causes frailty in patients and often prevents them from undergoing further therapies, and yet there is little effective therapy against cancer-associated cachexia[6]. Cancer patients with cachexia may have reduced food intake and metabolic dysregulation, including elevated energy expenditure, excess catabolism and inflammation but failed to be corrected by nutritional supplementation[7,8]. All these phenomena tightly correlate to adipose and skeletal muscle wasting, negative risk factors for cancer survival[9]. Elevations of factors including cytokines have been observed in cachectic patients, functioning as essential messengers to connect primary tumour with adipose tissue[10–12].

Adipose tissue plays important roles in regulating whole-body energy metabolism via its storage function in white adipose tissue (WAT) and its dissipating function in beige and brown adipose tissue

[1]Hubei Key Laboratory of Cell Homeostasis, College of Life Sciences, TaiKang Center for Life and Medical Sciences, Zhongnan Hospital of Wuhan University, Wuhan University, Wuhan, Hubei 430072, China. [2]Department of Endocrinology, Zhongnan Hospital of Wuhan University, Wuhan, Hubei 430062, China. [3]Department of Breast and Thyroid Surgery, Renmin Hospital of Wuhan University, Wuhan, Hubei 430060, China. [4]Hubei Key Laboratory of Cell Homeostasis, College of Life Sciences; TaiKang Center for Life and Medical Sciences; The Institute for Advanced Studies; Frontier Science Center for Immunology and Metabolism, Wuhan University, Wuhan, Hubei 430072, China. [5]These authors contributed equally: Yong Hu, Liu Liu. ✉e-mail: xjl100901@whu.edu.cn; weiyan@whu.edu.cn

(BAT). WAT typically contributes to energy accumulation, while BAT functions on the dissipation of energy as heat[13]. Additionally, WAT could turn to function like BAT in cancer patients, known as white adipose browning[14]. Recognised as an endocrine organ, adipose tissue not only responds to different signals from traditional hormone systems and the central nervous system, but also expresses and secretes factors with important endocrine functions including leptin[15]. Leptin plays an important role in regulating weight and energy consumption[16,17], and has been implicated in stress response, metabolic diseases, neurological disorders and cancer as a cytokine[18,19]. Adipocytes-derived leptin regulates gene expression related to cancer progression, such as adhesion, invasion, angiogenesis, signal transduction and apoptosis[20,21]. The level of leptin circulation in the blood is proportional to body mass index (BMI) and total fat storage in the body[22,23]. Beyond linking brain-adipose tissue crosstalk[18], leptin has also been reported to promote fat loss by activating the release of catecholamine signalling molecules from neurons wrapped around the fat cells[24,25]. Additionally, the elevation of leptin signal in the whole body has been related to breast cancer malignancy. It shapes the tumour microenvironment, mainly through its ability to potentiate both migrations of endothelial cells and angiogenesis[26]. However, whether and how the leptin signalling pathway is affected by primary BC tumour to exert systemic regulation is still unknown yet.

Tumour cells could communicate and influence distant organs, such as skeletal muscle, and islets, through secreting sEVs[27,28]. sEVs could encapsulate various bioactive cargos, including DNA, RNA, lipids and metabolites to exert systemic regulation[29–31]. Those remote organs or tissues are not usually colonised by cancer cells but the underlying mechanism is still poorly understood[32,33]. Whether cancer cell-derived sEVs are widely involved in the communication between primary tumour and WAT is still largely unknown. Here, we identify that BC-derived miR-204-5p functions as critical media during the crosstalk between primary tumour cells and white adipose tissue. This leads to lipolysis and energy expenditure with subsequent fat loss in cancer-associated cachexia. Collectively, our study reveals a dynamic interaction between cancer and host at the systemic level, and highlights the potential application of targeting both miR-204-5p and leptin signalling pathways in preventive screenings and early intervention for BC cachexia patients.

## Results

### Breast cancer-derived sEVs induced fat loss in white adipose tissue

To explore the effect of breast cancer (BC)-derived sEVs on adipose tissue, we xenografted engineered mouse triple-negative BC (TNBC) cell line 4T1 (4T1/Ctrl) and exosomes secretion impaired 4T1 cells (4T1/Rab27a KO) and those cells were stably expressing a membrane-targeted Lck-GFP (Fig. 1a). Both Rab27a knockdown in human TNBC MDA-MB-231 cells (231/Rab27a KD) and Rab27a knockout in 4T1 (4T1/Rab27a KO) cells with low to no detectable Rab27A expression, showed severely impaired capability for exosomes secretion (Supplementary Fig. 1a, b). We detected robust GFP signals within both epigonadal white adipose tissue (eWAT) and inguinal white adipose tissue (iWAT) from 4T1/Ctrl mice but much weaker signals in 4T1/Rab27a KO mice (Fig. 1b, c). Simultaneously, MDA-MB-231 cells and 231/Rab27a KD cells were used to establish orthotopic xenograft tumours in the mammary fat pad. Both the 4T1/Ctrl mice and 231/Ctrl mice demonstrated hypermetabolism, with obviously elevated oxygen consumption rate than either tumour free mice, 4T1/Rab27a KO or 231/Rab27a KD mice (Fig. 1d). Accordingly, the heat production rate was higher in 4T1/Ctrl mice and 231/Ctrl mice than their respective control groups (Fig. 1e). 4T1/Ctrl and 231/Ctrl mice lost ~6.41% and ~7.57% weight after five weeks and exhibited less food intake (Supplementary Fig. 1c, d). Strikingly, 4T1/Ctrl mice displayed much shrinking eWAT and iWAT morphology than tumour-free or 4T1/Rab27a KO mice and lower

eWAT, iWAT, gastrocnemius (GA) and tibial anterior (TA) weight were detected (Supplementary Fig. 1e, f). Oil Red O staining further evidenced there was much more fat loss in 4T1/Ctrl mice than their respective control groups (Supplementary Fig. 1g). All those observations suggested the potential possibility of cancer-derived sEVs involved in fat loss during cancer-associated cachexia.

To further confirm whether BC-derived sEVs participated in this regulation, female NOD/SCID/IL2Rγ-null (NSG) mice were treated with sEVs through semi-weekly intravenous injections for five weeks (Fig. 1f). MDA-MB-231 and non-cancer-mammary epithelial MCF-10A cells derived sEVs were endowed with classic exosomal marker-tetraspanin CD9 expression (Supplementary Fig. 1h). Immuno-fluorescence examination also demonstrated both MCF-10A sEVs and MDA-MB-231 sEVs could be uptake by iWAT and eWAT tissue in vivo (Fig. 1g, h). Further fluorescence showed more GFP signals were detected in both eWAT and iWAT from sEVs injected mice than PBS group (Fig. 1i). 4T1 sEVs were intravenously injected into female BALB/c mice semi-weekly for 5 weeks. With the assistance of micro-CT examination, we found the whole body fat deposition was much lower in mice receiving MDA-MB-231 sEVs and 4T1 sEVs than their respective control groups (Fig. 1j). MDA-MB-231 sEVs and 4T1 sEVs mice lost more weight over the course of five weeks and had less food intake compared respective control mice (Supplementary Fig. 1i, l). Similarly, there was obvious thinner iWAT and shrinking eWAT morphology in mice receiving MDA-MB-231 sEVs and 4T1 sEVs (Supplementary Fig. 1j). Significant mass loss was found in both eWAT, iWAT rather than BAT, GA or TA from mice receiving MDA-MB-231 sEVs and 4T1 sEVs when compared to their respective control groups (Supplementary Fig. 1k). Mice receiving MDA-MB-231 sEVs and 4T1 sEVs demonstrated hypermetabolism, with obviously elevated oxygen consumption rate (Fig. 1k). Accordingly, the heat production rate was much higher in mice receiving MDA-MB-231 sEVs and 4T1 sEVs when compared with their respective control groups (Fig. 1l). Oil Red O staining further evidenced remarkable reduction of fat accumulation in both iWAT and eWAT from mice receiving MDA-MB-231 sEVs than their respective control groups (Supplementary Fig. 1m). All the above sEVs injection data in vivo further evidenced cancer derived sEVs directly participated in the process of cancer-associated cachexia.

### Cancer-derived miR-204 promotes hypoxia signalling by targeting VHL in white adipose tissue

We next performed transcriptome sequencing for eWAT tissue from mice xenografted with MDA-MB-231 cells (231/Ctrl) and tumour-free control mice, and identified that hypoxia-inducible factor-1 (HIF1) and hypoxia metagene pathway were both upregulated in adipose tissue from 231/Ctrl mice (Fig. 2a). Subsequent qPCR with reverse transcription (qRT−PCR) examination validated the significant suppression for *Vhl* expression, while no significant alteration for *Hif1α* level was observed in eWAT and iWAT from 231/Ctrl mice when compared to tumour free mice or 231/Rab27a KD mice (Fig. 2b). Consistently, the protein abundance of VHL was significantly suppressed but HIF1A expression was remarkably induced in both eWAT and iWAT from 231/Ctrl than their relative control groups (Fig. 2c). Similar pattern was detected in mice receiving MDA-MB-231 sEVs when compared to mice receiving PBS or MCF-10A sEVs (Figs. 2d, e). To seek the underlying mechanism, we referred to the previously reported small RNA sequencing in MDA-MB-231 sEVs versus MCF-10A sEVs[34], and identified miR-204-5p (hereafter as miR-204) overlappingly enriched in predicted candidates pool potentially targeting *VHL* (Supplementary Table 1) (Fig. 2f). To explore the regulation of *VHL* by miR-204, we overexpressed miR-204 gene in MCF-10A cells (10A/miR-204) and found there was 15-fold upregulation of miR-204 expression in 10A/miR-204 cells derived sEVs when compared to MCF-10A sEVs (Supplementary Fig. 2a). Interrogation of sequences identified miR-204

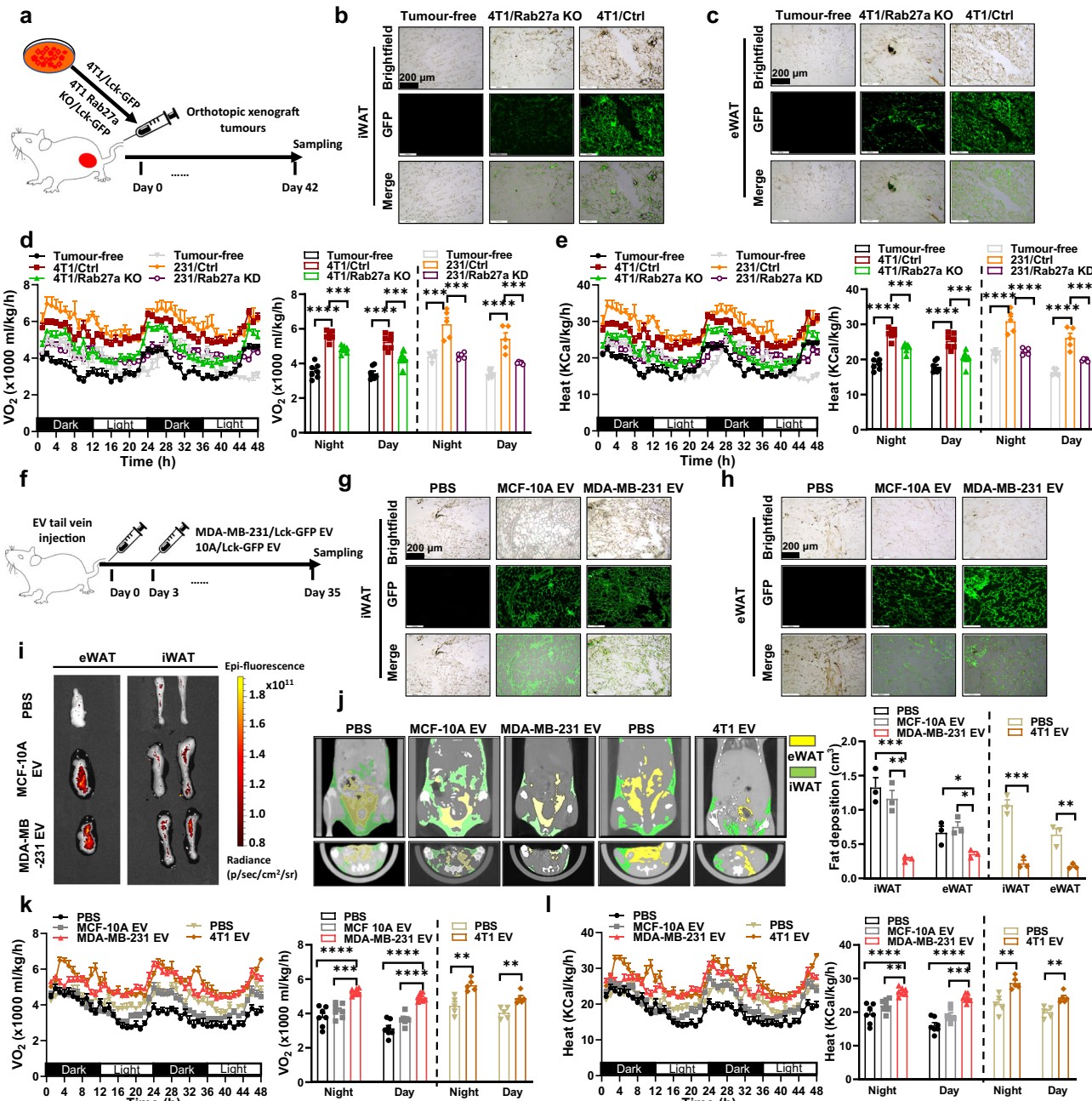

**Fig. 1 | Breast cancer-derived sEVs induces fat loss in white adipose tissue. a** 4T1/Lck-GFP cells and 4T1 Rab27a KO/Lck-GFP cells were used to establish orthotopic tumours in the BALB/c mice mammary fat pad. The schematic model was created using Adobe Illustrator. **b**, **c** GFP signals (green) in inguinal white adipose tissue (iWAT) /epigonadal white adipose tissue (eWAT) of bearing Lck-GFP tumours. **d**, **e** MDA-MB-231 cells, 231/Rab27a KD cells were injected into female NOD/SCID/IL2Rγ-null (NSG) mice mammary fat pad to form tumours. Oxygen consumption (VO₂) and heat production were monitored over a 48 h period in tumour group mice. Dot plots represent the average values for 48 h time window. $n = 7$ for BALB/c mice per group and $n = 5$ for NSG mice per group; One-way ANOVA followed by Dunnett multiple comparison test. **f** NSG mice were tail vein injected with ~10 µg sEVs per injection per mouse (MCF-10A/Lck-GFP sEVs, MDA-MB-231/Lck-GFP sEVs) twice a week. The schematic model was created was created using Adobe Illustrator. **g**, **h** GFP signals (green) in iWAT /eWAT of mice intravenously injected with Lck-GFP labelled sEVs. Repeated three times independently with similar results

obtained. Scale bar, 200 µm. **i** Ex vivo iWAT/eWAT in mice receiving Lck-GFP labelled sEVs. Red fluorescence levels in the iWAT and eWAT were acquired after 6 h with an IVIS Spectrum system. **j** BALB/c mice were tail vein injected with ~10 µg sEVs per injection per mouse (4T1 sEVs) twice a week. eWAT (yellow) and iWAT (green) distribution in NSG mice and BALB/c mice receiving sEVs, as visualised by micro-CT. ($n = 3$ per group; one-way ANOVA followed by Dunnett multiple comparison test). **k** Oxygen consumption (VO₂) and heat production **l** were monitored over a 48 h period in sEVs injected group mice. Dot plots represent the 48 h average values. For the body weight changes in NSG mice ($n = 7$ per group; one-way ANOVA followed by Dunnett multiple comparison test): 8.72 ± 4.37% (PBS group), 7.03 ± 0.63% (MCF-10A sEVs group) and −7.48 ± 6.95% (MDA-MB-231 sEVs group); For the body weight changes in BALB/c mice ($n = 5$ per group): 16.32 ± 4.52% (PBS group) and −2.35 ± 1.43% (4T1 sEVs group). Data are presented as mean±s.e.m; *$P < 0.05$, **$P < 0.01$,***$P < 0.001$, ****$P < 0.0001$. Source data and exact $P$ value are provided as a Source data file.

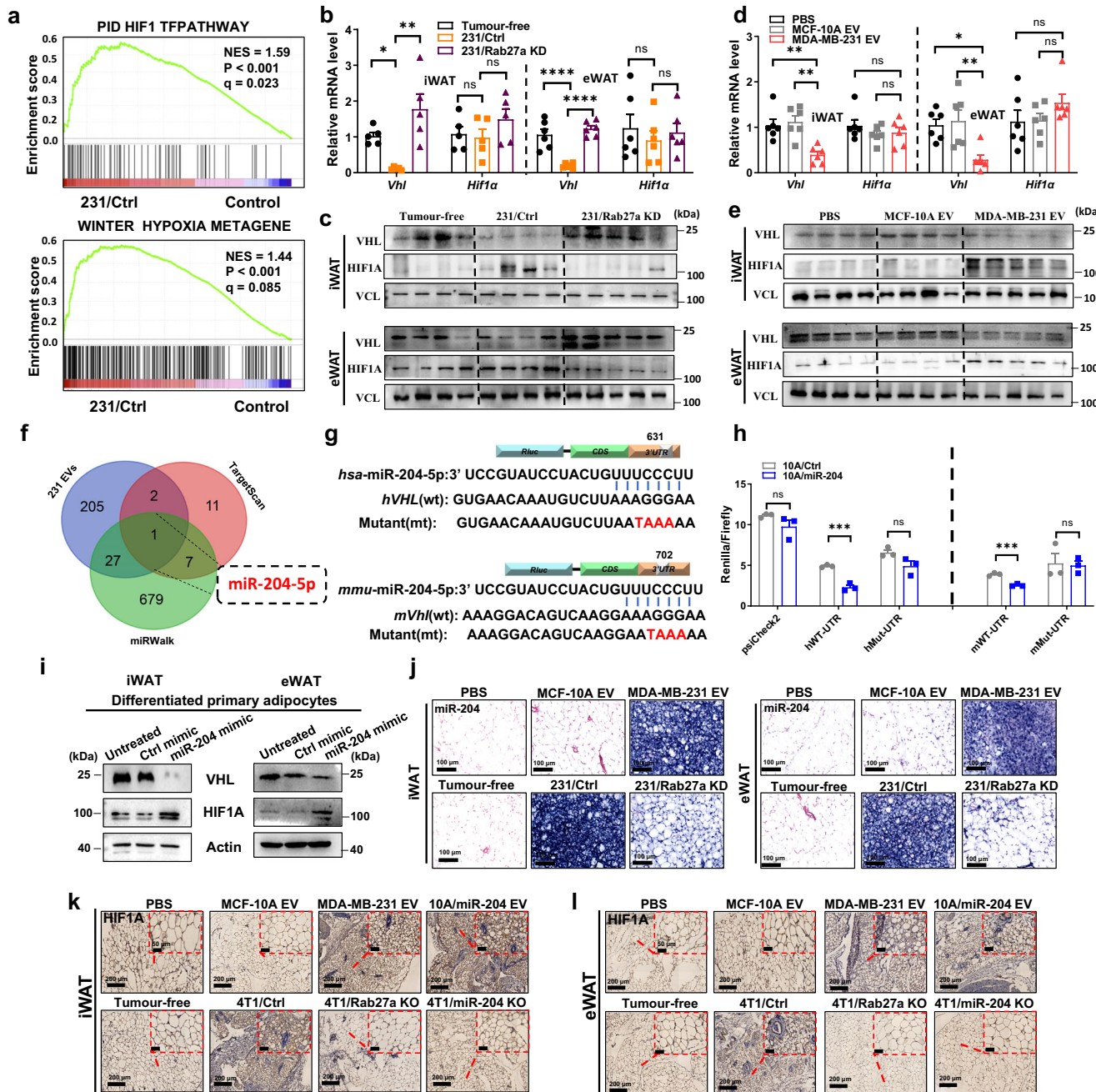

**Fig. 2 | Cancer cell-secreted miR-204 induces hypoxia by targeting VHL. a** The eWAT collected from orthotopic tumour mice were subjected to RNA-seq and GSEA analysis, showing enrichment of genes related to hypoxia pathways (*n* = 3 mice for Control, *n* = 3 mice for bearing MDA-MB-231 tumour). NES: normalized enrichment score. Nominal *P*-values and false discovery rate (FDR) *q*-values are shown. **b** Validation by qRT-PCR analysis of mRNA abundance of *Vhl* and *Hif1α* in iWAT and eWAT from mice bearing indicated tumours (*n* = 5 mice per group; one-way ANOVA followed by Dunnett multiple comparison test). **c** Immunoblot analysis of VHL and HIF1A in iWAT and eWAT from tumour group mice (tumour-free group, *n* = 4; 231/Ctrl group, *n* = 4; 231/Rab27a KD group, *n* = 5). Vinculin (VCL) was used as loading control. **d** Validation by qRT-PCR analysis of mRNA abundance of *Vhl* and *Hif1α* in iWAT and eWAT from mice in sEVs injected groups (*n* = 6 mice per group; one-way ANOVA followed by Dunnett multiple comparison test). **e** Immunoblot analysis of VHL and HIF1A in iWAT and eWAT from sEVs treated mice (PBS group, *n* = 4; MCF-10A sEVs group, *n* = 4; MDA-MB-231 sEVs group, *n* = 5). **f** Target gene

prediction of miR-204-5p with miRNA-seq analysis and related bioinformatics tools. **g** Predicted miR-204 binding sites in human and mouse *VHL/Vhl* genes (*hVHL* and *mVhl*, respectively). The corresponding sequences in WT and mutated reporters are shown. **h** Responsiveness of the reporters to miR-204 in transfected in MCF-10A (unpaired two-tailed *t*-test, *n* = 3 biological replicates). **i** Western blots showing levels of indicated proteins in differentiated primary adipocytes transfected with miR-204 mimic or control miRNA mimic. Repeated three times independently with similar results obtained. **j** Representative ISH images showing miR-204 level in adipose tissue of indicated groups of mice. Repeated three times independently with similar results obtained. Scale bar, 100 μm. Representative immunohistochemistry staining of HIF1A in iWAT and eWAT from xenografted tumour mice **k** and sEVs injected mice **l** and their respective control groups. Repeated three times independently with similar results obtained. Scale bar, 200 μm. Data are presented as mean ± s.e.m; *P* < 0.05, **P* < 0.01, ***P* < 0.001, ****P* < 0.0001, ns: not significant. Source data and exact *P* value are provided as a Source data file.

binding sites within the 3'UTR of both human and mouse *VHL/Vhl* genes. (Fig. 2g). Luciferase reporter assays comparing miR-204 responsiveness of wild-type and site-mutated constructs confirmed direct targeting 3' UTR of *VHL* by miR-204 through each predicted binding site in human and mouse, respectively (Fig. 2h). Transfection of miR-204 mimic inhibited VHL but induced HIF1A expression in differentiated primary adipocytes (Fig. 2i).

Consistently, in situ hybridisation (ISH) staining evidenced more miR-204 expression in both iWAT and eWAT of mice receiving MDA-MB-231 sEVs and 231/Ctrl mice when compared to their respective control groups (Fig. 2j). Induced HIF1A but adversely reduced VHL abundance was measured in both eWAT and iWAT from mice receiving 10A/miR-204 sEVs than MCF-10A sEVs (Supplementary Fig. 2b). In agreement with tissue effect, highly upregulated HIF1A protein rather than RNA level, and remarkably reduced VHL expression was detected in fractioned primary adipocytes of iWAT or eWAT from mice receiving MDA-MB-231 sEVs or 10A/miR-204 sEVs than PBS or MCF-10A sEV treated mice (Supplementary Fig. 2c, d). We also demonstrated mice receiving 10A/miR-204 sEVs endowed with less whole body weight, sectioned fat and skeletal muscle weight (Supplementary Fig. 2e, f), along with increased oxygen consumption rate than mice receiving MCF-10A sEVs but lower fat deposition (Supplementary Fig. 2g, h). Additionally, we found 5–10 fold of upregulation for miR-204 abundance in iWAT and eWAT from the mice receiving high miR-204 sEVs or 4T1/Ctrl mice than their relative control groups (Supplementary Fig. 2i). To examine whether miR-204 regulated the adipose tissue in vivo, we generated miR-204 KO cells by employing CRISPR-Cas9 technology (Supplementary Fig. 2j). There was 10-fold of plasma miR-204 enrichment in 10A/miR-204 sEVs mice and 6-fold tumour-bearing mice than their relative control mice (Supplementary Fig. 2i). Reduced HIF1A but conversely elevated VHL abundance was measured in both eWAT and iWAT from 4T1/miR-204 KO mice comparing tumour bearing mice (Supplementary Fig. 2k). Meanwhile, immunohistochemistry (IHC) staining showed the protein level of HIF1A was highly elevated in both iWAT and eWAT from 4T1/Ctrl mice when compared with 4T1/miR-204 KO, 4T1/Rab27a KO or tumour free mice (Fig. 2k). Similar trends were detected in both iWAT and eWAT from mice receiving high miR-204 sEVs (MDA-MB-231 sEVs and 10A/miR-204 sEVs) when compared to their respective control groups (Fig. 2l). Intriguingly, 4T1/miR-204 KO mice gained more body and fat weight, respectively (Supplementary Fig. 2l, m). MCF-10A sEVs and 4T1/miR-204 KO mice displayed lower oxygen consumption rate but restored food intake capability than 10A/miR-204 sEVs and 4T1/Ctrl mice (Supplementary Fig. 2n, o). All the findings indicated circulating miR-204 was closely related to hypermetabolism and energy consumption in vivo.

## Cancer cells derived miR-204 induced leptin signalling pathway in white adipose tissue

When we performed GSEA enrichment analysis for our RNA sequencing (RNA-seq) on eWAT from mice that received MDA-MB-231-derived sEVs and 231/Ctrl mice and their related control mice, we identified signalling by leptin was highly upregulated in MDA-MB-231 sEVs and 231/Ctrl mice when compared to the control group (Fig. 3a). Further qRT-PCR showed *leptin* was highly induced in eWAT and iWAT from mice receiving 10A/miR-204 sEVs or tumour-bearig mice compared to their respective control groups (Fig. 3b, d). Consistently, the protein abundance of LEPTIN was highly induced in eWAT and iWAT from mice receiving 10A/miR-204 sEVs or tumour-bearing mice compared to their respective control groups (Fig. 3c, e). Immunohistochemical analyses of adipose tissues showed increased LEPTIN expression in iWAT and eWAT from mice receiving 10A/miR-204 sEVs, MDA-MB-231 sEVs and 4T1/Ctrl mice, with reduced adipocyte size, a multilocular lipid droplet phenotype when compared to their respective control groups (Fig. 3f, g). Enzyme-linked immunosorbent assay (ELISA)

quantification also confirmed leptin was significantly elevated in both adipose tissue and serum from mice receiving 10A/miR-204 sEVs, MDA-MB-231 sEVs and 4T1/Ctrl mice when compared to their respective control groups (Fig. 3h). When we treated 3T3-L1 and stromal vascular fraction (SVF) cells with carboxyfluorescein succinimidyl ester (CFSE)-labelled MDA-MB-231 sEVs or 10A/miR-204 sEVs, we detected sEVs were efficiently uptake by adipocytes in vitro (Supplementary Fig. 3a, b). Consistently, elevated leptin concentration was detected in cultured medium of SVF cells with MDA-MB-231 sEVs and 10A/miR-204 sEVs treatment than that of the cells incubated with PBS or MCF-10A sEVs under normoxia rather than hypoxia condition (Supplementary Fig. 3c).

To examine whether hypoxia induced leptin expression in our model, we knocked down HIF1A and found the expression of LEPTIN was inhibited simultaneously under hypoxia condition (Supplementary Fig. 3d). Conversely, LEPTIN was induced by a degradation-resistant form of HIF1A (dPA), HIF1A construct carrying a double proline to alanine substitution (dPA) that prevents hydroxylation and ubiquitin-proteasomal degradation even under normoxia (Supplementary Fig. 3e). Indeed, miR-204 mimic had no effect on the VHL mediated LEPTIN expression under hypoxia (Supplementary Fig. 3f). High miR-204 sEVs educated SVF cells enhanced the expression of HIF1A under normoxia but alleviated the regulation of LEPTIN under hypoxia due to exogenous HIF1A expression (Supplementary Fig. 3g). When we added SVF cells with $CoCl_2$ to induce hypoxia, we detected overexpression of HIF1A induced LEPTIN expression, while the pattern that hypoxia erased all the regulatory effect of 10A/miR-204 sEVs or MDA-MB-231 sEVs in differentiated primary adipocytes from both iWAT and eWAT (Supplementary Fig. 3h). Inversely, when we overexpressed VHL to suppress the expression of HIF1A in 3T3-L1 cells, the abundance of HIF1A was sharply reduced and lost the responsiveness by miR-204 sEVs (Fig. 3i). Interestingly, there was no significant change for fatty acid accumulation among SVF cells incubation with MDA-MB-231 sEVs, 10A/miR-204 sEVs, MCF-10A sEVs or PBS in vitro (Supplementary Fig. 3i), indicating the WAT browning needs systemic regulation mediated by leptin connecting WAT and brain in vivo.

Considering leptin signals proopiomelanocortin (POMC) neurons of the hypothalamus through the hypothalamic signal transducer and activator of transcription 3 (STAT3) pathway to regulate appetite[35], we observed elevated hypothalamic STAT3 phosphorylation level induced by high miR-204 sEVs-injected mice and tumour-bearing mice than their respective control mice (Fig. 3j). High miR-204 sEVs treatment or tumour-bearing mice indeed reduced mRNA abundance of agouti-related neuropeptide (*Agrp*) and neuropeptide Y (*Npy*) in hypothalamic, respectively, thus regulating energy balance and neuroendocrine function (Fig. 3k). Hence, we examined the concentration and cellular origin of norepinephrine (NE) in iWAT and eWAT. Compared with control mice, NE concentrations were much higher in the iWAT of mice receiving 10A/miR-204 sEVs or MDA-MB-231 sEVs as well as elevated NE levels in iWAT from 4T1/Ctrl mice than their respective control groups (Fig. 3l).

## Circulating miR-204 promotes thermogenesis and lipolysis in white adipose tissue

Considering leptin-NE signals through β3-adrenergic receptors induce PKA signalling to promote lipolysis and thermogenesis[35], we found mice receiving MDA-MB-231 sEVs and 10A/miR-204 sEVs endowed with higher body temperature than respective control mice upon acute cold exposure (Fig. 4a). Consistent pattern was detected in 4T1/Ctrl versus tumour-free, 4T1/miR-204 KO or 4T1/Rab27a KO mice (Fig. 4b). Genes related to thermogenesis were highly induced in iWAT and BAT receiving high miR-204 sEVs or 4T1/Ctrl mice than their respective control groups (Fig. 4c, d). Meanwhile, western blot showed the expression of thermogenesis and lipolysis-related proteins were

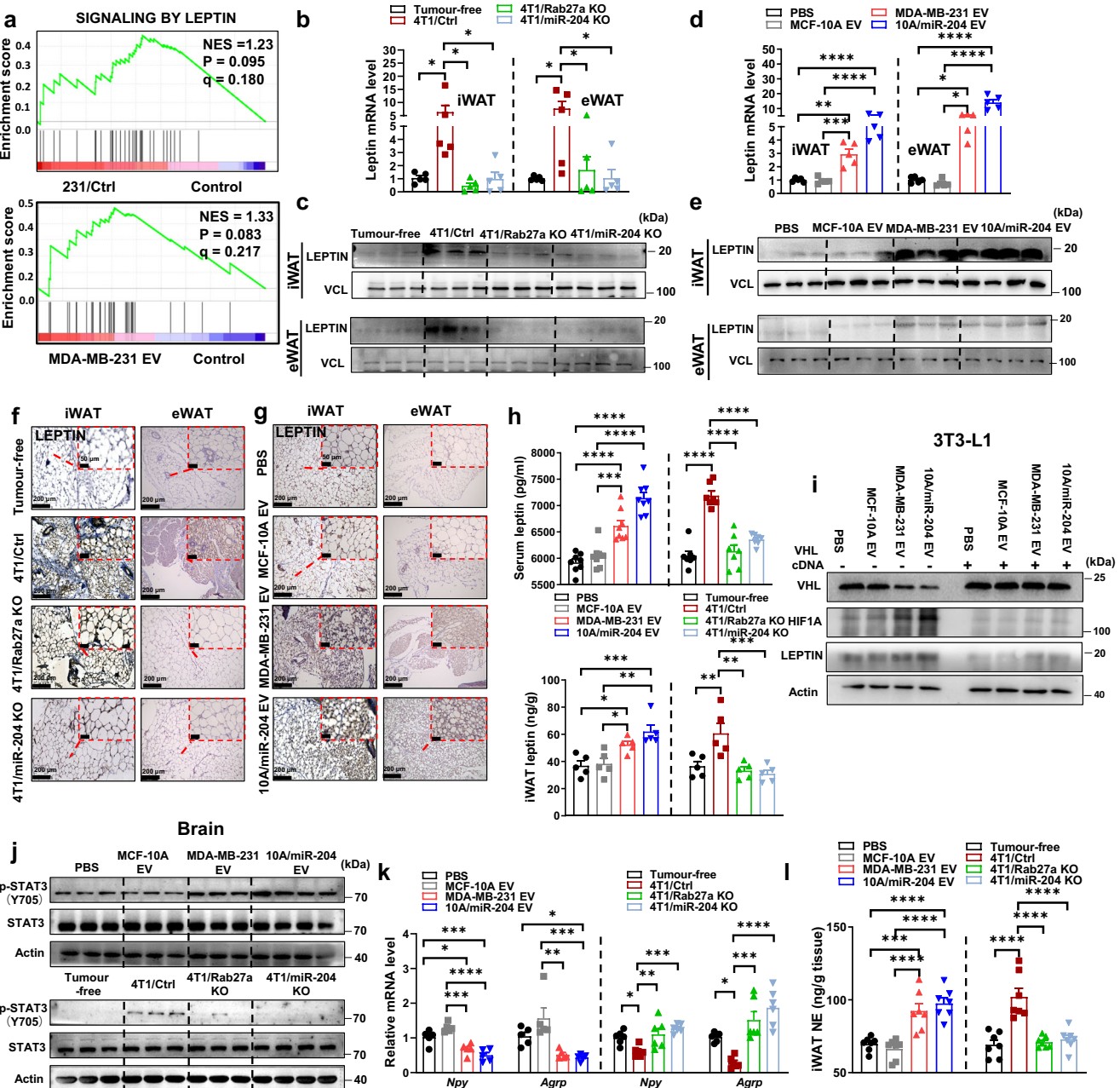

**Fig. 3 | Circulating miR-204 induces leptin signalling pathway in white adipose tissue. a** RNA-seq and GSEA showing enrichment of genes related to leptin pathways (*n* = 3 mice for Control; *n* = 3 mice for MDA-MB-231 sEVs; *n* = 3 for 231/Ctrl mice). Nominal *P* values and FDR *q* values are shown. **b** Validation by qRT-PCR analysis of mRNA abundance of *leptin* in iWAT and eWAT from mice bearing indicated tumours (*n* = 5 mice per group). **c** Immunoblot analysis of leptin in iWAT and eWAT from tumour group mice (tumour-free group, *n* = 3; 4T1/Ctrl group, *n* = 3; 4T1/Rab27a KO group, *n* = 3; 4T1/miR-204 KO group, *n* = 4). **d** Validation by qRT-PCR analysis of mRNA abundance of *leptin* in iWAT and eWAT from mice in sEVs injected groups (*n* = 5 mice per group). **e** Immunoblot analysis of leptin in iWAT and eWAT from sEVs treated mice (PBS group, *n* = 3; MCF-10A sEVs group, *n* = 3; MDA-MB-231 sEVs group, *n* = 3; 10 A/miR-204 sEVs group, *n* = 4). **f, g** Representative immunohistochemistry staining of leptin in iWAT and eWAT from each group mice. Scale bar, 200 μm. **h** Leptin levels of serum (*n* = 8 for sEVs injection mice; *n* = 7 for

mice bearing indicated tumours) and iWAT (*n* = 5 mice per group) were examined with ELISA kit. **i** Western blots showing indicated proteins in mature 3T3-L1 transfected with Vhl cDNA plasmid with or without indicated sEVs treatment. Repeated three times independently with similar results obtained. **j** Immunoblot analysis of STAT3 phosphorylation at Tyr705 in hypothalamus of various experimental group mice. (PBS group, *n* = 3; MCF-10A sEVs group, *n* = 3; MDA-MB-231 sEVs group, *n* = 3; 10 A/miR-204 sEVs group, *n* = 4; tumour-free group, *n* = 3; 4T1/Ctrl group, *n* = 3; 4T1/Rab27a KO group, *n* = 3; 4T1/miR-204 KO group, *n* = 4). **k** The qRT-PCR analysis of mRNA abundance of *Agrp*, *Npy* in hypothalamus (*n* = 5 for sEVs injection mice; *n* = 6 for BALB/c mice bearing 4T1-derived tumours). **l** Levels of norepinephrine (NE) in iWAT tested by ELISA (*n* = 7 mice per group). Data are presented as mean ±s.e.m; *P* < 0.05, **P* < 0.01, ***P* < 0.001, ****P* < 0.0001, one-way ANOVA followed by Dunnett multiple comparison test was used for **b**, **d**, **h**, **k**, **l**. Source data and exact *P* value are provided as a Source data file.

altered by miR-204 (Fig. 4e, f). There was more plasma-free fatty acids level but sharply reduced triglyceride accumulation in the iWAT from mice receiving high miR-204 sEVs and 4T1/Ctrl mice than their respective control group mice (Fig. 4g, h). Moreover, primary white adipocytes extracted from 10A/miR-204 sEV or MDA-MB-231 sEV-treated mice displayed a higher oxygen consumption rate but lower glycolytic rate than PBS or MCF-10A sEVs treated mice adipocytes (Fig. 4i, j). Intriguingly, primary white adipocytes from mice injected with

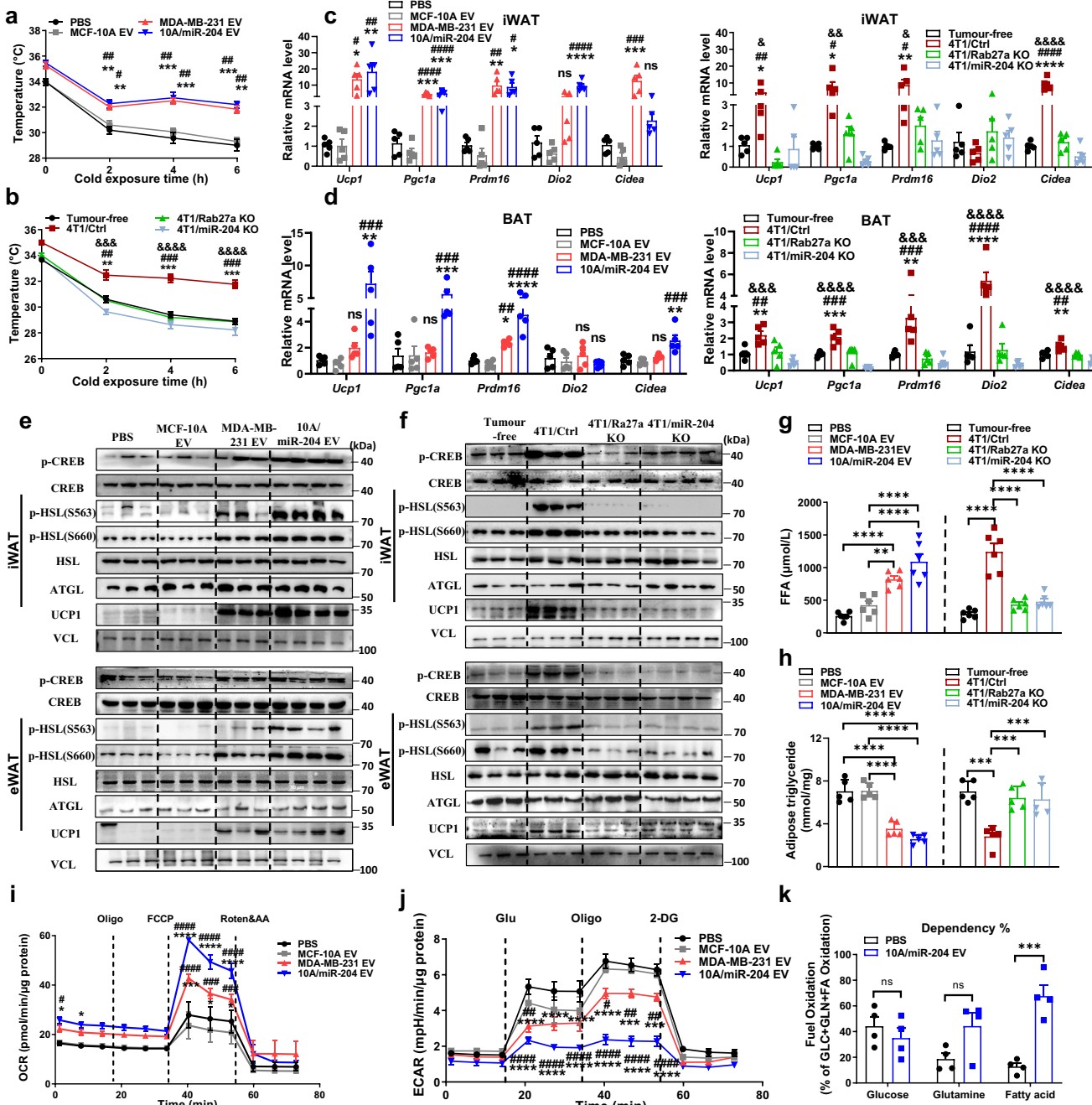

**Fig. 4 | Circulating miR-204 promotes thermogenesis and lipolysis in white adipose tissue. a** Rectal temperatures were monitored during acute cold challenge for 6 h (4 °C) of the sEVs injected group mice, n = 3 mice per group. One-way ANOVA followed by Dunnett multiple comparison test. **b** Rectal temperatures were monitored during acute cold challenge for 6 h (4 °C) of mice xenografted with tumour, n = 3 mice per group. One-way ANOVA followed by Dunnett multiple comparison test. **c** Relative mRNA abundance of the thermogenic markers in iWAT from sEVs treated mice and tumour-bearing group mice (n = 5 mice per group; one-way ANOVA followed by Dunnett multiple comparison test). **d** Relative mRNA abundance of the indicated thermogenic markers in BAT from sEVs treated mice and tumour-bearing group mice (n = 5 mice per group; one-way ANOVA followed by Dunnett multiple comparison test). **e** Western blots showing levels of UCP1 and indicated proteins in iWAT/eWAT of sEVs injection groups of mice (PBS group, n = 3; MCF-10A sEVs group, n = 3; MDA-MB-231 sEVs group, n = 3; 10 A/miR-204 sEVs group, n = 4) and **f** tumour bearing mice and their respective control groups (tumour-free group, n = 3; 4T1/Ctrl group, n = 3; 4T1/Rab27a KO group, n = 3; 4T1/ miR-204 KO group, n = 4). **g** Serum free fatty acid levels in the various experimental group (n = 6 mice per group; one-way ANOVA followed by Dunnett multiple

comparison test). **h** Triglyceride (TG) content of iWAT in the various experimental group (n = 5 mice per group; one-way ANOVA followed by Dunnett multiple comparison test). **i** Oxygen consumption rate (OCR) and **j** extracellular acidification rate (ECAR) in primary white adipocytes isolated from iWAT of each group mice, including basal respiration, uncoupled respiration (by stimulating uncoupling with FCCP) and nonmitochondrial respiration (with rotenone) was determined using Seahorse metabolic analyser. Real time triplicate readings (upper panel) and their averages (lower panel) were shown (n = 4; two-way ANOVA followed by Dunnett multiple comparison test). The ECAR was evaluated after the addition of 10 mM glucose. **k** The dependence of mitochondrial substrate in primary white adipocytes isolated from iWAT of mice receiving 10 A/miR-204 sEVs or PBS group determined by XF Mito fuel flex test (n = 4 mice per group; unpaired two-tailed t-test). Data are presented as mean±s.e.m. *P < 0.05, **P < 0.01, ***P < 0.001, ****P < 0.0001 compared with PBS or tumour-free mice; #P < 0.05, ##P < 0.01, ###P < 0.001, ####P < 0.0001 compared with MCF-10A EV or 4T1/Rab27a KO; &P < 0.05, &&P < 0.01, &&&P < 0.001, &&&&P < 0.0001 compared with 4T1/miR-204 KO in **a–d**, **i**, **j**; ns: not significant. Source data and exact P value are provided as a Source data file.

10A/miR-204 sEVs exhibited higher dependency on fatty acids as an energy source (Fig. 4k).

To examine whether the effect of circulating miR-204 on thermogenesis and lipolysis-mediated leptin signalling pathway, we employed the db/db mice lack leptin receptor to examine the effect of cancer-derived miR-204 on thermogenesis and lipolysis in WAT. Although C57BL/6 mice treated with 10A/miR-204 sEVs exhibited lower body weight and fat mass compared to the control group mice, db/db mice displayed no significant whole body weight or fat mass difference upon 10A/miR-204 sEVs injection (Supplementary Fig. 4a, b). Similarly, db/db mice displayed mild changes of food intake and cold exposure upon miR-204 sEVs treatment (Supplementary Fig. 4c, d). Consistently, elevated oxygen consumption and heat production was observed in C57BL/6 mice, but db/db mice had mild oxygen consumption alteration in mice receiving miR-204 sEVs when compared with MCF-10A sEVs treated mice as well (Supplementary Fig. 4e, f). No significant regulation of either leptin induced by miR-204 sEVs in eWAT and iWAT tissue of db/db mice (Supplementary Fig. 4g, h). Likewise, db/db mice received 10A/miR-204 sEVs displayed undetectable basal and phosphorylation of hormone-sensitive lipase (HSL) alteration and fat deposition in both eWAT and iWAT (Supplementary Fig. 4i, j). Additionally, micro-CT showed decreased fat in C57BL/6 mice but not db/db mice(Supplementary Fig. 4k), and H&E staining showed no significant lipid droplet size change in db/db mice receiving miR-204 sEVs (Supplementary Fig. 4l). Although miR-204 containing sEVs did induce HIF1A and LEPTIN expression and suppressed VHL expression in WAT of db/db mice (Supplementary Fig. 4g–i), the metabolic effects were missing in these mice lacking the leptin signalling. These results indicated miR-204 exerted the regulation on the white adipose tissue through the leptin signalling pathway.

## Circulating miR-204 altered mitochondria content in white adipose tissue

GSEA enrichment also showed electron transport chain oxphos system in mitochondria pathway was highly induced in 231/Ctrl eWAT than control mice, with clusters of respiring genes induction (Fig. 5a, b). Taking the ratio of mitochondrially encoded cytochrome c oxidase 1 (*Mt-Co1*)/NADH:ubiquinone oxidoreductase core subunit V1 (*Ndufv1*) in mitochondria as the internal reference, the content of mitochondria in adipose tissue was detected by RT-qPCR. The content of mitochondria in 4T1/Ctrl mice and mice receiving high 10A/miR-204 sEVs was ~2-fold when compared to their respective control groups (Fig. 5c, d). Moreover, electron microscopy analysis showed tumour-derived miR-204 sEVs induced reduction in lipid droplet size in differentiated primary adipocytes from miR-204 sEV-treated mice, indicating that miR-204 enhanced lipid mobilisation to fuel adipocyte thermogenesis and altered the morphology and quantity of mitochondria in mouse adipose tissue and heat production increased upon sEVs injection (Fig. 5e). H&E and immunohistochemistry staining of adipose tissues showed distinct signatures of multilocular, beige-like adipocytes and higher intensity of UCP1 protein signals in iWAT and eWAT from sEVs group mice and 4T1/Ctrl group mice compared to their respective control (Fig. 5f–i).

Although CCK8 revealed 4T1/miR-204 KO had no influence on cancer cells proliferation (Supplementary Fig. 5a), the tumour volumes, tumour weight and Ki67 positive cells from mice xenografted with 4T1 tumours were much larger than 4T1/miR-204 KO tumours in vivo (Supplementary Fig. 5b, c, d). To study the metabolic reprogramming of circulating miR-204 mediated lipolysis in adipose tissue, we performed lipid metabolomics analysis and detected an enrichment of phosphatidylethanolamine in serum from 4T1 tumour-bearing mice than 4T1/miR-204 KO mice and sEVs educated adipocytes in vitro (Supplementary Fig. 5e–h). PE (18:2-19:1) significantly stimulated the proliferation, migration and invasion of 4T1 and MDA-MB-231 cancer cells in a dose-dependent manner (Supplementary Fig. 5i, j),

suggesting the pro-tumour effect of PE mediated by miR-204/*VHL* mediated leptin signalling pathway.

## Exogenous VHL expression suppresses adipose tissue browning

To finally determine to what degree BC tumours dysregulate adipose tissue through miR-204/VHL axis, we constructed adeno-associated virus 9 (AAV9) expressing mouse *Vhl* cDNA and injected the AAV-VHL (TB AAV-VHL) into adipose tissue following BC tumour implantation, with AAV-GFP (TB AAV-GFP) served as a control. GFP signals were detected in adipose tissue more than in other organs or xenograft tumour of AAV-GFP-injected mice (Supplementary Fig. 6a). Both iWAT and eWAT mass were increased upon VHL restoration along with slightly increased body weight (Fig. 6a, b). There was reduced tumour weight and growth rate in TB AAV-VHL mice when compared to TB AAV-GFP, as well as remarkably reduced Ki67 expression within TB AAV-VHL tumours (Supplementary Fig. 6c, d). TB AAV-VHL mice exhibited more food intake than TB AAV-GFP mice and displayed lower body temperature (Fig. 6c, d). Notably, TB AAV-GFP mice lost ~10.11% body weight (w/o tumor) over the course of the experiment. Besides, energy expenditure and the expression of BAT thermogenic genes were also reversed under the overexpression of VHL (Fig. 6e–g). There was a higher trend of respiratory exchange ratio (RER) while a lower trend of locomotor activity in the TB AAV-GFP group rather than in TB AAV-VHL (Supplementary Fig. 6e, f).

TB AAV-VHL effectively restored VHL and decreased leptin expression in adipocytes of tumour-bearing mice, and reversed tumour-associated adipose tissue lipolysis and browning. Exogerous VHL expression effectively suppresses the expression of *leptin, β3-AR*, uncoupling protein 1 (*Ucp1*) and PR domain containing 16 (*Prdm16*) in both eWAT and iWAT from TB AAV-VHL than TB AAV-GFP mice (Fig. 6h). We found that VHL was successfully overexpressed in the eWAT and iWAT of TB AAV-VHL mice, along with decreased HIF1A and LEPTIN, and reversed lipolysis related protein expression (Fig. 6i). Immunohistochemical analyses also confirmed the reverse expression pattern of VHL and LEPTIN in both eWAT and iWAT between TB AAV-GFP and TB AAV-VHL (Supplementary Fig. 6g). TB AAV-VHL mice had the reduced plasma leptin and adipose tissue resided NE level when compared to TB AAV-GFP mice (Fig. 6j, k). Similar decreased levels of plasma-free fatty acid accumulation in TB AAV-VHL mice when compared with TB AAV-GFP mice (Fig. 6l). H&E staining of iWAT and eWAT revealed that TB AAV-VHL mice were endowed with much larger lipid droplets size than TB AAV-GFP mice (Fig. 6m). Immunohistochemical analyses of adipose tissues and western blot of their derived SVF cells showed TB AAV-VHL restored the expression of LEPTIN, HIF1A and UCP1 (Fig. 6n, o, Supplementary Fig. 6h). These data demonstrated that adipocytes overexpressing VHL could block miR-204 sEVs induced lipolysis and browning, leading to protection against tumour associated cachexia.

## Circulating miR-204 promotes white adipose tissue lipolysis in colon and lung carcinoma

To identify whether the regulation of thermogenesis and lipolysis by circulating miR-204 universally existed in other cancer models with cachexia, we employed two established murine syngeneic graft models of cachexia, colon carcinoma (xenografted with C26 cells, C26/Ctrl), and Lewis lung carcinoma (xenografted with LLC cells, LLC/Ctrl) with miR-204 KO in each cell lines (Fig. 7a and Supplementary Fig. 7a). Tumour-bearing mice from both cancer models lost more weight over the course of five weeks than the control groups (Fig. 7b and Supplementary Fig. 7b). Indeed, the weights of iWAT, epididymal white adipose tissue (eWAT), GA and TA were significantly lower in C26/Ctrl and LLC/Ctrl group mice than their respective controls (Fig. 7c and Supplementary Fig. 7c). C26/Ctrl and LLC/Ctrl mice exhibited less food intake than respective miR-204 KO mice and displayed higher body temperature upon acute cold exposure (Fig. 7d, e and Supplementary

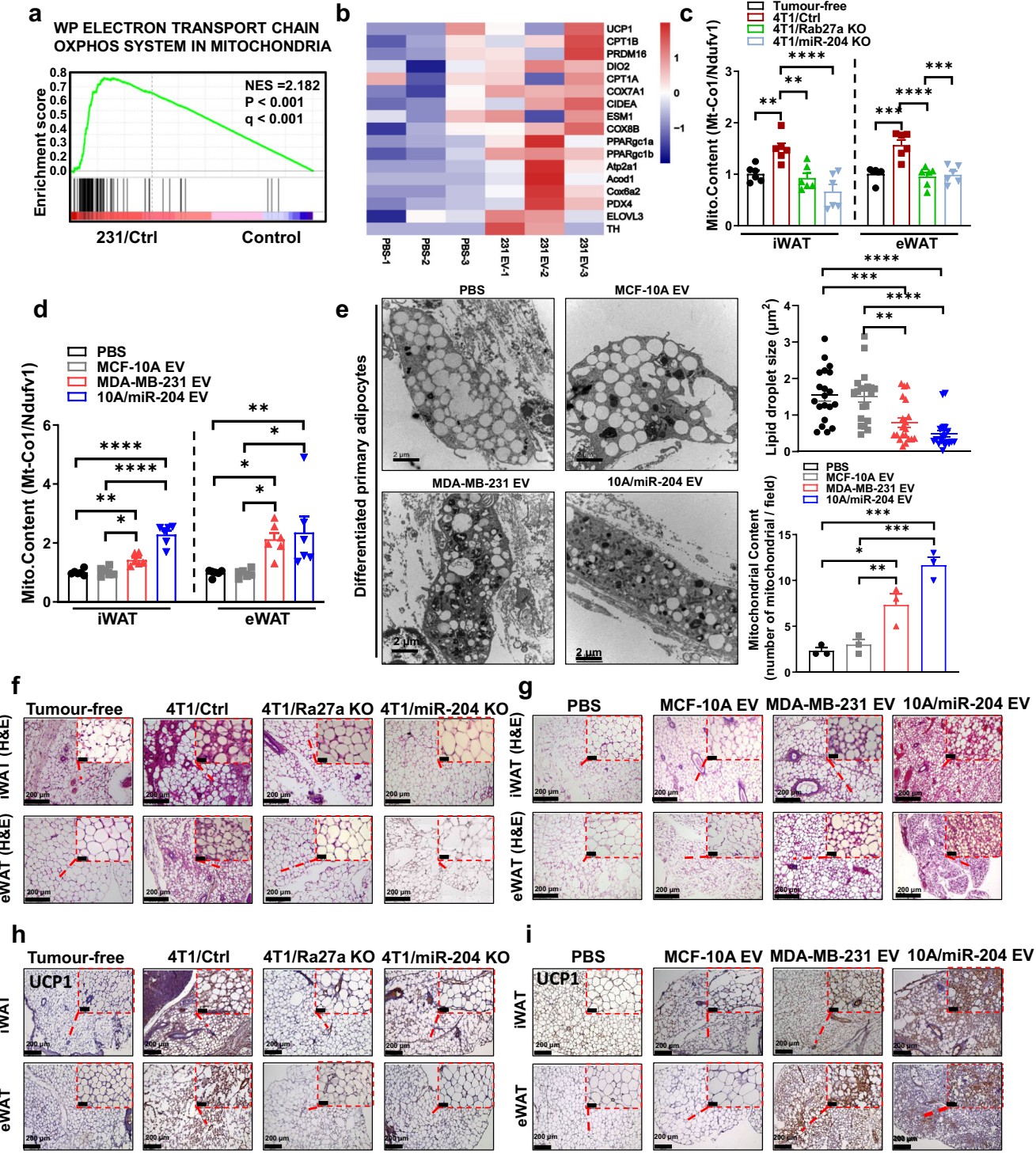

**Fig. 5 | Circulating miR-204 alters mitochondria content in white adipose tissue. a** GSEA showing enrichment of genes related to indicated pathways (*n* = 3 mice for Control and *n* = 3 mice for 231/Ctrl mice). Nominal *P* values and FDR *q* values are shown. **b** Heat maps showing thermogenic gene expression in eWAT from mice receiving PBS and MDA-MB-231 sEVs. **c**, **d** The qRT-PCR analysis of mRNA abundance of *Mt-Co1/Ndufv1* in iWAT and eWAT (*n* = 6 mice per group; one-way ANOVA followed by Dunnett multiple comparison test). **e** Representative transmission electron microscopy (TEM) images (left), quantification of lipid droplet size and mitochondrial content (right) in differentiated primary adipocytes from iWAT

following sEVs treatment (*n* = 3 random fields per group; one-way ANOVA followed by Dunnett multiple comparison test). Representative H&E staining with iWAT and eWAT in mice from xenograft tumour model **f** and mice receiving sEVs injection model **g**. Repeated three times independently with similar results obtained. Scale bar, 200 μm. Representative immunohistochemistry staining of UCP1 in iWAT and eWAT from both xenograft tumour model **h** and mice receiving sEV injection model **i** Repeated three times independently with similar results obtained. Scale bar, 200 μm. Data are presented as mean ± s.e.m; *$P$ < 0.05, **$P$ < 0.01, ***$P$ < 0.001, ****$P$ < 0.0001. Source data and exact *P* value are provided as a Source data file.

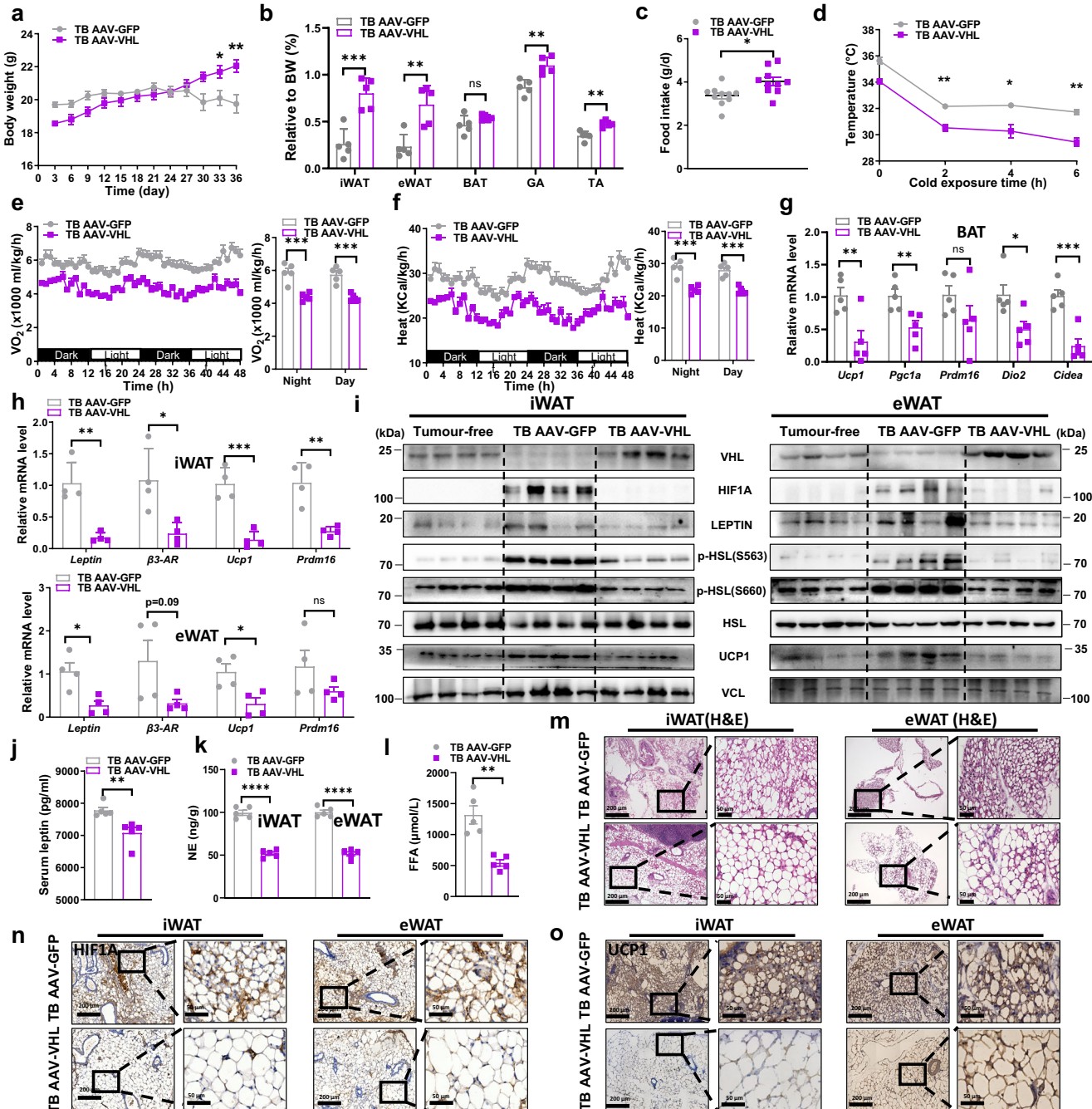

**Fig. 6 | Exogenous VHL expression suppresses white adipose tissue lipolysis and browning.** AAV-VHL viruses or AAV-GFP viruses were injected into iWAT and eWAT of both sides 1 week after implantation of 4T1 cells. Organs were collected on day 42 after implantation. All the parameters were collected from AAV-VHL and AAV-GFP mice. **a** Body weight ($n = 5$ mice per group; unpaired two-tailed $t$-test). **b** Relative fat tissues weight (normalized to body weight) ($n = 5$ mice per group; unpaired two-tailed $t$-test). **c** Daily food intake ($n = 5$ mice examined by 10 independent experiments per group; unpaired two-tailed $t$-test). **d** Rectal temperatures of the mice at different times after cold exposure ($n = 3$ mice per group; unpaired two-tailed $t$-test). **e** Oxygen consumption (VO$_2$) and **f** Heat production. Dot plots represent the 48-hour average values ($n = 5$ mice per group; unpaired two-tailed $t$-test). **g** Relative mRNA abundance of the indicated thermogenic markers in BAT from each group mice ($n = 5$ mice per group; unpaired two-tailed $t$-test). **h** Relative mRNA abundance of the indicated mRNA levels in iWAT and eWAT from each group mice ($n = 4$ mice per group; unpaired two-tailed $t$-test). **i** Western blots

showing indicated proteins in iWAT and eWAT from each group mice (tumour free group, $n = 4$ mice per group; TB AAV-GFP group, $n = 4$ mice per group; TB AAV-VHL group, $n = 4$ mice per group). **j** The serum leptin levels ($n = 5$ mice per group) and **k** level of norepinephrine (NE) in iWAT and eWAT ($n = 5$ mice per group), unpaired two-tailed $t$-test. **l** Plasma-free fatty acid levels in the two group ($n = 5$ mice per group; unpaired two-tailed $t$-test). **m** Representative H&E staining with iWAT/eWAT from each group mice. Repeated three times independently with similar results obtained. Scale bar, 200 μm. **n** Representative immunohistochemistry staining of HIF1A in iWAT and eWAT. Repeated three times independently with similar results obtained. Scale bar, 200 μm. **o** Representative immunohistochemistry staining of UCP1 in iWAT and eWAT. Repeated three times independently with similar results obtained. Scale bar, 200 μm. Data are presented as mean ± s.e.m; $^*P < 0.05$, $^{**}P < 0.01$, $^{***}P < 0.001$, $^{****}P < 0.0001$, ns: not significant. Source data and exact $P$ value are provided as a Source data file.

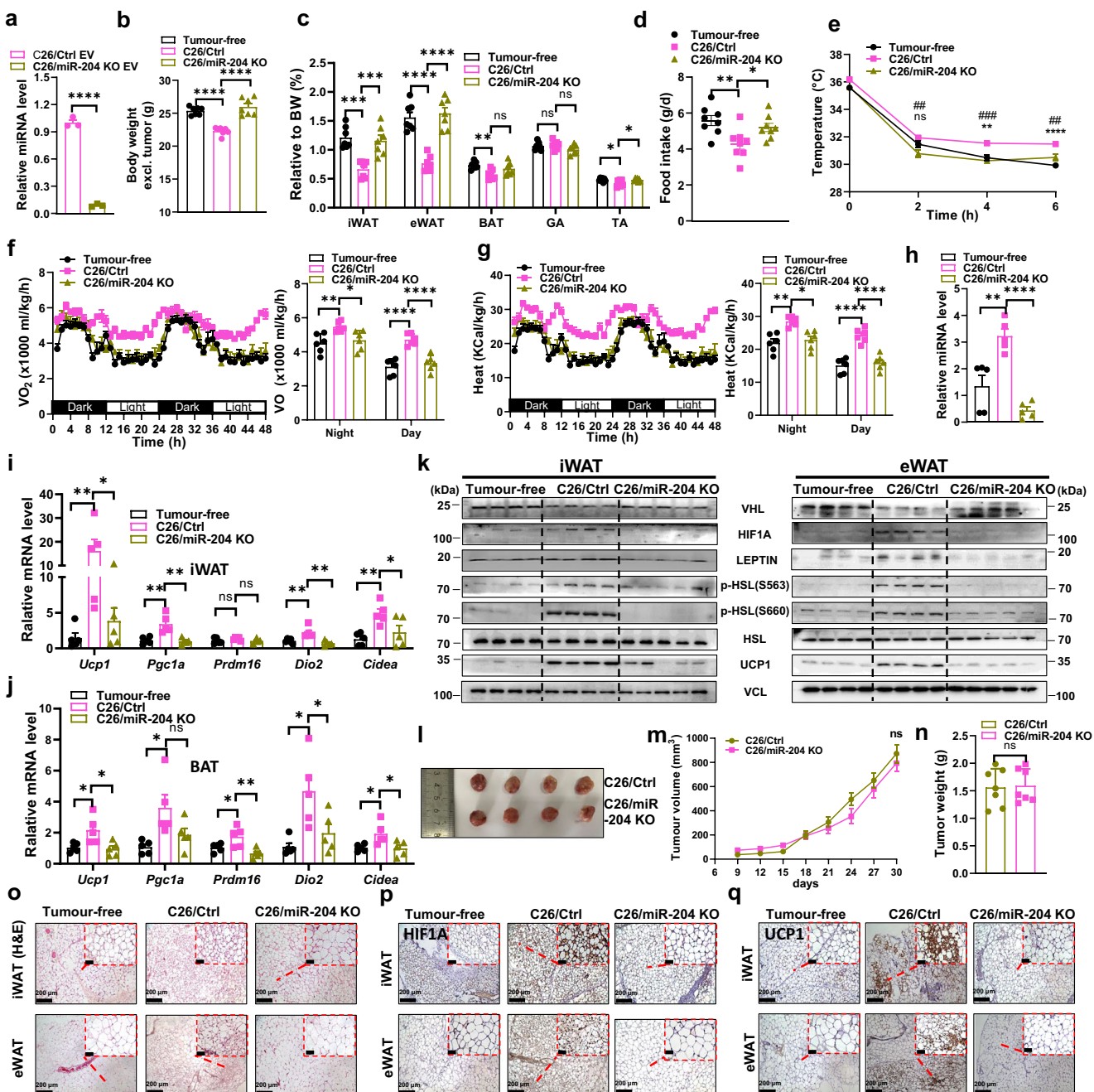

**Fig. 7 | Exosomal miR-204 promotes thermogenesis and lipolysis in C26-xenografted tumour mice. a** The expression level of miR-204 in C26 and C26/miR-204 KO derived sEVs (*n* = 3 biologically independent samples per group; unpaired two-tailed *t*-test). **b** Body weight (calculated by substracting tumour weight from the total weight) of mice bearing C26 tumours (*n* = 7 mice per group; one-way ANOVA followed by Dunnett multiple comparison test). **c** Relative tissues weight included iWAT, epididymal white adipose tissue (eWAT), BAT, GA, TA (normalized to body weight) in tumour-free, C26/Ctrl and C26/miR-204 KO group mice (*n* = 7 mice per group; one-way ANOVA followed by Dunnett multiple comparison test). **d** Daily food intake (*n* = 8 mice per group; one-way ANOVA followed by Dunnett multiple comparison test). **e** Rectal temperatures of the mice at different times after cold exposure (*n* = 5 mice per group; one-way ANOVA followed by Dunnett multiple comparison test). **f** The oxygen consumption (VO₂) and **g** Heat production from indicated groups. Dot plots represent the 48 h average values (*n* = 6 mice per group; one-way ANOVA followed by Dunnett multiple comparison test). **h** The qRT-PCR analysis of miR-204 abundance of serum sEVs from each group mice (*n* = 5 mice per group; one-way ANOVA followed by Dunnett multiple comparison test).

**i**, **j** Relative mRNA abundance of the indicated thermogenic markers in iWAT and BAT from each group mice (*n* = 5 mice per group; one-way ANOVA followed by Dunnett multiple comparison test). **k** Western blots showing indicated proteins in iWAT and eWAT from each group mice. (tumour free group, *n* = 4 mice per group; C26/Ctrl group, *n* = 4 mice per group; C26/miR-204 KO group, *n* = 5 mice per group). **l** Image of excised tumours (*n* = 4 mice per group). **m** Tumour volumes were measured every 3 days (n = 7 mice per group; unpaired two-tailed *t*-test) and (**n**) tumour weight when mice were sacrificed (*n* = 7 mice per group; unpaired two-tailed *t*-test). **o** Representative H&E staining with iWAT/eWAT from each group mice. Repeated three times independently with similar results obtained. Scale bar, 200 μm. **p** Representative immunohistochemistry staining of HIF1A in iWAT and eWAT. Repeated three times independently with similar results obtained. Scale bar, 200 μm. **q** Representative immunohistochemistry staining of UCP1 in iWAT and eWAT. Repeated three times independently with similar results obtained. Scale bar, 200 μm. Data are presented as mean±s.e.m; *P < 0.05, **P < 0.01, ***P < 0.001, ****P < 0.0001, ns: not significant. Source data and exact P value are provided as a Source data file.

Fig. 7d, e). Meanwhile, tumour-bearing mice in both models demonstrated hypermetabolism, and the rate of oxygen consumption and heat product was significantly elevated (Fig. 7f, g and Supplementary Fig. 7f, g). We then examined the serum exosomal miR-204 level and detected elevated miR-204 in C26/Ctrl and LLC/Ctrl group mice compared with respective controls (Fig. 7h and Supplementary Fig. 7h). Both iWAT and BAT exhibited elevated expression of thermogenic genes *Ucp1, Pgc1a, Prdm16, Dio2, Cidea* and lipolysis related proteins (Fig. 7i–k and Supplementary Fig. 7i–k). Interestingly, C26 tumour induced adipose tissue wasting without any effect on the tumour volume or tumour mass when compared with C26/miR-204 KO group (Fig. 7l–n). In comparison, LLC tumour promoted adipose tissue lipolysis compared with LLC/miR-204 KO tumour accompanied by smaller tumour volumes (Supplementary Fig. 7l–n). Histological and immunohistochemical analyses of adipose tissues showed clear signs of browning in iWAT and eWAT of LLC/Ctrl and C26/Ctrl rather than C26/miR-204 KO and LLC/miR-204 KO tumour-bearing mice, with reduced adipocyte size, a multilocular lipid droplet phenotype, and increased HIF1A, UCP1 protein abundance (Fig. 7o–q and Supplementary Fig. 7o–q).

## Exosomal miR-204 is associated with leptin signalling in BC patients

We tested a panel of human and mouse BC cell lines and their derived sEVs, which all demonstrated a higher abundance of miR-204 in various BC cell lines when compared with normal breast mammary epithelial MCF-10A cells and MCF-10A sEVs fraction, suggesting higher circulating miR-204-5p levels might universally distribute among distinct BC types (Fig. 8a, b). In addition, we also tested colon cancer, lung cancer, and gastric cancer cells, which showed significantly higher abundance of miR-204-5p in C26, LLC and MGC-803 cell lines and their derived sEVs (Fig. 8c–e). Nanoflow cytometry analysis, iodixanol density gradient ultracentrifugation and transmission electron microscope (TEM) further confirmed the typical size, density and markers of exosomes (Supplementary Fig. 8a–c). sEVs from these BC cells, rather than MCF-10A cells, suppressed VHL and corresponding regulation of LEPTIN and HIF1A in human adipocytes (Fig. 8f). Similarly, high miR-204 sEVs treated human adipocytes displayed higher HIF1A expression but suppressed VHL level (Fig. 8g, h). We included an additional group of six non-cancer control females that were selected to match the cachexia patients with BC to assess the function of serum-derived sEVs, we purified total circulating sEVs from BC patients and non-cancer controls, and observed miR-204 was higher in the BC group (Fig. 8i). The level of miR-204 was much higher in BC tissue derived sEVs as well (Fig. 8j). We next analysed serum samples from eighteen female patients with BC who had lost weight in 6 months, together with a group of eighteen non-cancer control females that had never been diagnosed with cancer and were initially selected to match the age and BMI of patients with BC. Sera from the BC group showed higher concentration of LEPTIN (Fig. 8k). Accordingly, we found higher miR-204 expression especially in cachexia patients when compared to BC patients with obesity (Fig. 8l). There was much lower VHL level but endowed with higher HIF1A, LEPTIN, UCP1 and phosphorylated HSL protein expression in tumour associated adipose tissue as well (Fig. 8m, n). Collectively, in our model, BC-derived miR-204 induced cancer-associated cachexia by reprogramming hypoxia and leptin signalling pathway to trigger lipolysis by targeting Vhl in white adipose tissue (Fig. 8o).

## Discussion

Tumour secretome functions as a pro-cachexia driving force to directly elicit catabolism in target tissues such as adipose tissue, skeletal muscle and heart[33,36]. Those complex molecules are majorly some inflammatory cytokines including tumour necrosis factor-alpha (TNF-α), interleukin 6 (IL-6) and the TGF-β family members, as well as

parathyroid hormone-like peptide (PTHrP) and prostaglandins which are considered to elicit the wasting disorder[37]. In our current model, we propose a communication mode mediated by miRNAs encapsulated by extracellular vesicles (EVs) between adipose tissue and primary tumour. As cancer-associated cachexia is characterised as a multi-organ wasting disorder with mass loss in both adipose tissue and skeletal muscle, we also observed the reduced skeletal muscle mass under miR-204-5p sEVs treatment but this effect was fully blocked in db/db mice (Supplementary Fig. 4b), suggesting the regulation of circulating miR-204 on skeletal muscle atrophy highly depends on the lipolysis of adipose tissue. Of note, the reduced muscle mass percentage is far less than that in adipose tissue. Specifically, we observed a ~1.0% decrease for iWAT and eWAT while only a 0.15% reduction for GA and TA muscle in 4T1 or MDA-MB-231 xenografted tumour mice. Parallelly, we found there was more fat loss than skeletal muscle mass loss in mice receiving MDA-MB-231 sEVs or 4T1 sEVs (Supplementary Fig. 1f, k). Therefore, the effect of BC-derived miR-204 on cachexia results more from the lipolysis of adipose tissue rather than muscle atrophy. Although the direct effect and mechanism of miR-204-5p on skeletal muscle has not been included in our study, it has been reported that miR-204-5p inhibits myoblast differentiation by targeting myocyte enhancer factor 2C (MEF2C) and estrogen-related receptor gamma (ERRγ)[38]. We have observed smaller tumour volumes and reduced metastatic capability in 4T1/miR-204 KO and MDA-MB-231/miR-204 KO mice when compared to their respective control groups (Supplementary Fig. 5b–d and 8d–f). However, in another C26 xenograft tumour model, we found C26/miR-204 KO group mice with similar tumour volume but endowed with different lipolysis when compared to wild type tumour bearing mice, suggesting weight loss may partially result from the lack of miR-204.

Actually, leptin production has been reported to be regulated by glucocorticoids, cytokines, agonists of peroxisome proliferator activated receptor gamma (PPARγ) and hypoxia, besides the sympathetic nervous system[39–45]. Suppression of exosomal miR-204 secretion or blocking regulation between miR-204/VHL axis in adipose tissue might hold promise for fighting cancer-associated cachexia and improving patient survival. However, sEVs containing high miR-204 failed to directly activate lipolysis in adipocytes, further supporting systemic regulation is required for the effect of miR-204. Besides, obesity drives changes in the levels of hormones such as leptin, adiponectin and ghrelin, which all have an immunomodulatory function, it remains critically important to determine whether and how circulating miR-204 intersects these metabolic, hormonal and immunologic regulations simultaneously during white adipose tissue browning and cancer-associated cachexia.

The plasma leptin levels are significantly higher in women than in men, even after the adjustment for total body fat mass[46]. Previous large cohort studies also reported elevated serum leptin levels in breast cancer patients[20]. The pro-tumoural effects of leptin are facilitated by its mitogen actions and leptin acts directly on tumour growth, migration and invasion signalling pathways or affects tissue insulin sensitivity, inflammatory responses and tumour angiogenesis to exert neoplastic effects in breast cancer[47,48]. Additionally, leptin could also increase hypothalamic mTOR activity, a cellular fuel sensor whose hypothalamic activity is directly tied to the regulation of energy intake[49]. Indeed, we observed food intake was significantly reduced in 4T1 xenograft-bearing mice or mice receiving MDA-MB-231-EVs than their respective control groups. The elevated leptin level could also directly inhibit appetite by regulating the activity of AgRP and POMC in the hypothalamic neurons of the brain[50]. Leptin exerts metabolic and thermogenic effects either by changing sympathetic neural activity or dynamically regulating the architecture of the sympathetic nervous structure in adipose tissue[35]. However, sympathetic neuron-derived norepinephrine is a classical activator of lipolysis, BAT activity, and WAT browning in response to cold exposure[18,25]. Consistently, clinical

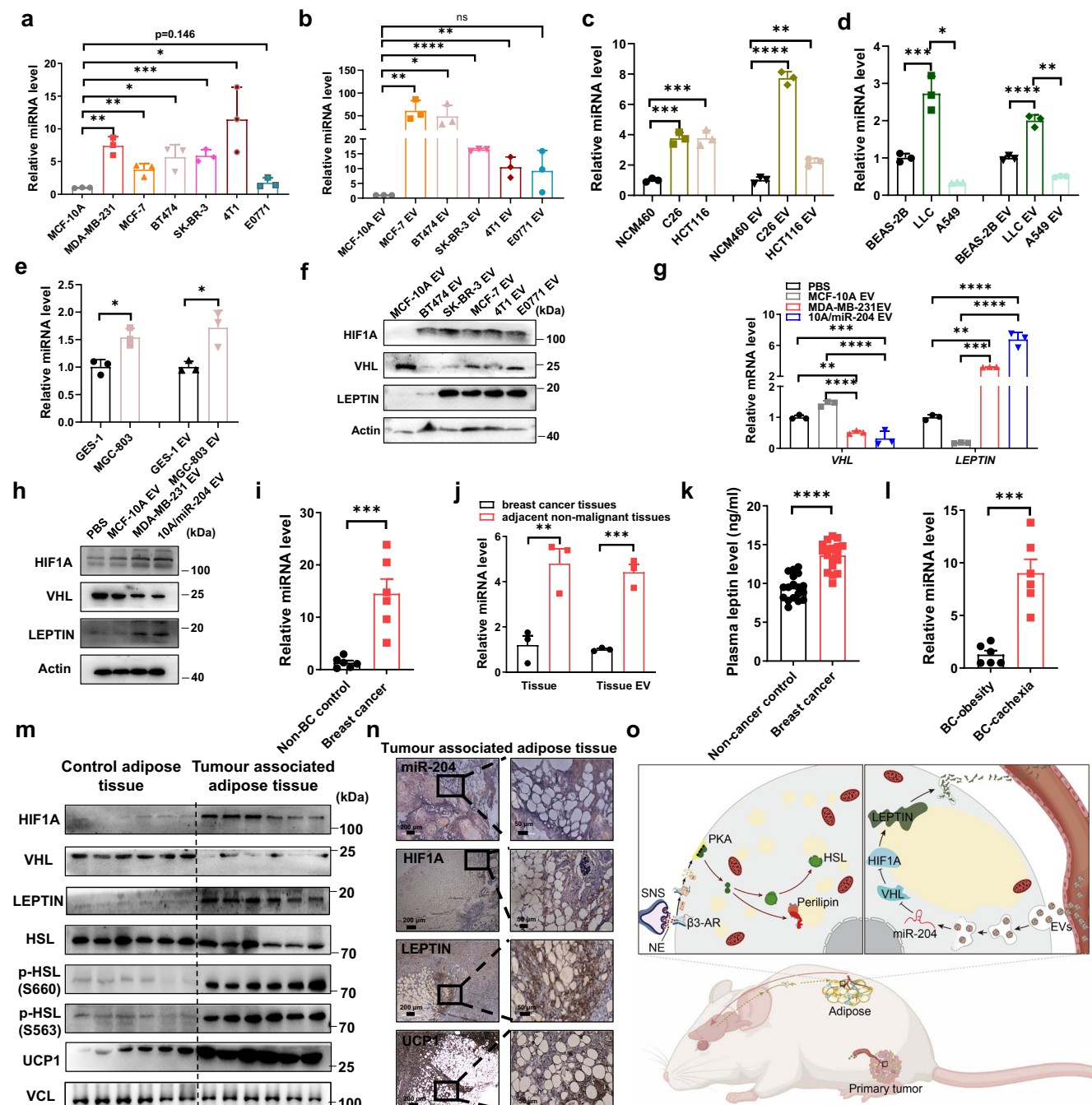

**Fig. 8 | Exosomal miR-204 is associated with leptin signaling in BC patients.** The qRT-PCR analysis of miR-204 abundance in various breast cancer cells ($n = 3$ biologically independent samples per group; unpaired two-tailed $t$-test) (**a**) and sEVs fraction ($n = 3$ biologically independent samples per group; unpaired two-tailed $t$-test). **b** The qRT-PCR analysis of miR-204 abundance in several cancer cell lines and their derived sEV fractions: (**c**) C26 and HCT116 colon cancer ($n = 3$ biologically independent samples per group; one-way ANOVA followed by Dunnett multiple comparison test), (**d**) LLC, A549 lung cancer cells ($n = 3$ biologically independent samples per group; one-way ANOVA followed by Dunnett multiple comparison test) and (**e**) MGC-803 gastric cancer cells ($n = 3$ biologically independent samples per group; unpaired two-tailed $t$-test). **f** Western blots showing indicated proteins in differentiated primary adipocytes treated various breast cancer sEVs. Repeated three times independently with similar results obtained. **g** Relative mRNA abundance of the indicated genes in human stromal vascular fraction (SVF) treated indicated sEVs ($n = 3$ biologically independent samples per group; one-way ANOVA followed by Dunnett multiple comparison test) and (**h**) the

related protein levels. **i** The qRT-PCR determined miR-204 levels in normal donor serum sEVs and breast cancer patient's serum sEVs ($n = 6$ per group; unpaired two-tailed $t$-test). **j** The qRT-PCR determined miR-204 levels in tumour tissue and sEVs ($n = 3$ biologically independent samples per group). **k** Serum leptin level in nomal donor and breast cancer patient ($n = 18$ per group). **l** The qRT-PCR determined miR-204 levels in BC patients with obesity and BC cachexia patient's serum sEVs ($n = 6$ per group; unpaired two-tailed $t$-test). **m** Western blots showing indicated proteins in control WAT from non-cancer patients or breast cancer patient's WAT. Repeated three times independently with similar results obtained. **n** Representative ISH images and immunohistochemistry staining of HIF1A, UCP1 and LEPTIN in tumour associated adipose tissues. Repeated three times independently with similar results obtained. Scale bar, 200 μm. **o** Schematic model was created using BioRender (https://biorender.com/) and Adobe Illustrator. Data are presented as mean±s.e.m; $^*P < 0.05$, $^{**}P < 0.01$, $^{***}P < 0.001$, $^{****}P < 0.0001$, ns: not significant. Source data and exact $P$ value are provided as a Source data file.

trials demonstrated that β-adrenergic blockade administration prevents loss of fat mass, attenuates WAT remodelling, and protects from WAT and body weight loss in advanced BC and the fact that β-adrenergic antagonism partially improved cachexia in patients affected with some of these diseases suggests a more general relevance of the mechanism described in our study[51]. Taken together, we show that sympathetic innervation supported by an extracellular vesicles-mediated neuroprotective environment is essential for WAT browning and adipose atrophy in BC. Targeting this intra-adipose communication axis between adipose and sympathetic neurons may provide a therapeutic option to treat cachexia.

## Methods

### Ethical statement
This research complies with all relevant ethical regulations. Specifically, human specimens were obtained from Renmin Hospital of Wuhan University (Wuhan, China), and approved by the Clinical Research Ethical Committee of Renmin Hospital of Wuhan University in accordance with the principle of the Helsinki Declaration. Informed consent from all participants was obtained. All animal experiments were handled under the Guidelines of the China Animal Welfare Legislation, and approved by the Institutional Animal Care and Use Committee at the College of Life Science, Wuhan University.

### Cells and constructs
Cell lines used in this study were obtained from American Type Culture Collection and cultured in the recommended media or as indicated. These include MDA-MB-231 (HTB-26), 4T1 (CRL-2539), SK-BR-3 (HTB-30), MCF-7 (HTB-22), E0771 (CRL-3461), MCF-10A (CRL-10317), BT474 (HTB-20), 3T3-L1 (CL-173), HEK 293T (ACS-4500), LLC (CRL-1642), NCM460 (CRL-9609), HCT116 (CCL-247), A549 (CCL-185). BEAS-2B was obtained from China Center for Type Culture Collection (GDC0139). C26 was gift from Dr. Pengcheng Bu of Institute of Biophysics, Chinese Academy of Sciences (Beijing, China). Both MGC-803 and GSE-1 cells were obtained from the Dr. Wenhua Li of Wuhan University (Wuhan, China). All cells used herein were tested to be free of mycoplasma contamination and authenticated by using the short tandem repeat profiling method. Mycoplasma detection was tested by PCR with Forward primer: 5′-GGGAGCAAACAGGATTAGATACCCT-3′ and Reverse primer: 5′-TGCACCATCTGTCACTCTGTTAACCTC-3′ on a MiniAmp thermocycler (Applied Biosystems, USA). 3T3-L1 cells were differentiated in the induction medium (DMEM/F12 medium containing 10% FBS, penicillin-streptomycin, and glutamine and then induced with a differentiation cocktail consisting of 0.5 mM 3-isobutyl-1-methylxanthine, 1 µM dexamethasone, 10 µg/mL insulin, 0.2 mM indomethacin, and 1 µM rosiglitazone in DMEM supplemented with 10% FBS, PS and glutamine) for 7–10 days[52]. Differentiation was confirmed by Oil Red O staining. When indicated, the primary adipocytes were cultured under 1% $O_2$ in an Anaerobic culture chamber (MAWORDE). MDA-MB-231/Lck-GFP and MCF-10A/Lck-GFP cells were constructed[53].

To construct 3T3-L1/Hif1α KD cells, short hairpin RNA targeting Hif1α sequence GGAAACTCCAAAGCCACTTCG and TGGATAGCGA-TATGGTCAATG was designed by BLOCK-iT RNAi Designer (Thermo Fisher Scientific) and the corresponding oligonucleotides were cloned into pLKO.1 puro vector (Addgene, 8453). A similar strategy was used to construct MDA-MB-231/Rab27a KD cells using two shRNAs (DNA sequences GCTGCCAATGGGACAAACATA and CCAGTGTACTTTACCA ATATA). To construct 10A/miR-204 cells, PCR primers 5′-TTACTC GAGTCCAGGGGAATCTTTGCTT and 5′-CGCGGATCCAGGAAGAAATA-CAACAGGGTGC were used to clone the sequence of human miR-204. The PCR-amplified fragments were digested with XhoI and BamHI and then inserted into the same sites of pLVX-IRES-ZsGreen1 vector. Then the plasmid was transfected into cells, and the GFP+ cells were sorted by flow cytometry. Cells were transduced by the lentivirus and selected in puromycin. 231/miR-204 KO, 4T1/miR-204 KO, C26/miR-204 KO and LLC/miR-204 KO cells were constructed using CRISPR-Cas9 genomic editing system. Single-guide RNAs (the combination of GAACTAT-TAGTCTTTGAGAG and GATGGTGGTTAGTTGCAAGA was selected for *hsa*-miR-204, and the combination of GCTAATGCTTTGTACGTGGA and CCAAAGTCATGGGCAGTTAC was selected for *mmu*-mir-204) predicted by the sgRNA Designer (https://portals.broadinstitute.org/gpp/public/analysis-tools/sgrna-design) were synthesised in DNA form and annealed into double strands, treated with T4 polynucleotide kinase and inserted into the BbsI-digested lentiCRISPR v2 vector (Addgene, 52961). The two constructs were co-transfected into cells, and then cells were screened for puromycin resistance, and the knockout efficiency of miR-204 knockout cell line was validated by qPCR. Monoclones were screened by genotyping PCR and confirmed by sequencing (Tsingke). 4T1/Rab27a KO cells using sgRNA (DNA sequence AGCGTCCCTGAAGAATGCAG). Monoclones were screened by western blot and confirmed by sequencing. To construct AAV-VHL, mouse *Vhl* cDNA was generated by PCR using primers 5′-TCCGAGCTCGCCACCATGCCCCGGAAGGCA and 5′-GATACCGGTT-CAAGGCTCCTCTTCCAGG. The resulting construct pscAAV-CMV-mouse *Vhl* was confirmed by sequencing (Tsingke). AAV serotype 9 was used to achieve robust transgene expression in adipose tissue. The 7 µg expression plasmid pscAAV-mouse Vhl, 7 µg pAAV2/9n (Addgene, 112865) and 20 µg pAdDeltaF6 (Addgene, 112867) were transfected into 15 cm dish HEK 293T cells with the assistance of Lipofectamine 2000 (Life Technologies), followed by the production of viruses. Titre was determined by qPCR (primers: 5′-CTCAGCCCTACCCGATCTTAC and 5′-ACATTGAGGGATGGCACAAAC). For mouse injection, $1.5 \times 10^{12}$ genome copies resuspended in 100 µL PBS was used. To construct reporters for miR-204-mediated regulation of VHL, a region encompassing partial coding region and 3′ UTR of human *VHL* gene (primers 5′-CCGCTCGAGTTCACTAGGCATTGTGAT and 5′-ATAGCGGCCGCAAT CAGGAATGTCACATA) and the 3′ UTR of mouse *Vhl* gene (primers 5′-CCGCTCGAGATTATGTCTCCTGACATT and 5′-ATAGCGGCCGCAAG TGTTAGGATGTAGTT) were cloned by PCR and inserted into the XhoI-NotI sites of psiCHECK-2 reporter vector (Promega) downstream of the Renilla luciferase (Rluc) gene. PCR primers 5′-TTAATAAAAATCA TTTTTGTAGGAAGCATT and 5′-TGATTTTTATTAAGACATTTGTTCAC TGTA (for mutated human miR-204 binding site), or 5′-AGGAA-TAAAAAGCAGAAAGGCTCTTATGTACTC and 5′- TGCTTTTTATTCC TTGACTGTCCTTTCTGTT (for mutated mouse miR-204 binding site) were used to clone mutants of VHL 3′ UTR. All plasmids were validated by sequencing. The *hsa*-miR-204-5p mimic (UUCCCUUUGUCAUCC UAUGCCU) was purchased from Ribo Life Sciences (miR10000265-1-5) and their corresponding negative controls (UUUGUACUACACAAAA-GUACUG) were purchased from Ribo Life Sciences (miR1N0000001-1-5) (Suzhou, China).

### Primary white adipocyte culture and fractionated primary adipocytes
Inguinal stromal vascular fraction (SVF) cells and epigonadal SVF cells were obtained from 30–35 days-old female mice by the following procedure. Fat tissue was dissected, washed with PBS, minced and digested for 45 min at 37 °C in PBS containing 10 mM $CaCl_2$, 1.5 U/mL collagenase I (BasalMedia, S361RV). Digested tissue was filtered through a 70 µm cell strainer and centrifuged at 600 g for 5 min to pellet the SVF cells. These were then resuspended in adipocyte culture medium (DMEM/F12(1:1); Gibco,12634028) plus glutamax, pen/strep (Gibco, 15140163), and 10% FBS (Sigma-Aldrich,12007 C), filtered through a 40 µm cell strainer, centrifuged as above, resuspended in adipocyte culture medium and plated. The SVF cells were grown to confluency for adipocyte differentiation, which was induced by the adipogenic cocktail containing 1 µM dexamethasone(Sigma-Aldrich, D4902), 5 µg/mL insulin (Sigma-Aldrich, I2643), 0.5 µM

isobutylmethylxanthine (Sigma-Aldrich, I5879), and 1 μM rosiglitazone (Sigma-Aldrich, R2408) in adipocyte culture medium. Two days after induction, cells were maintained in adipocyte culture medium containing 5 μg/mL insulin and 1 μM rosiglitazone. Medium was changed every two days. Starting at day 6, cells were maintained in adipocyte culture medium only and treated with various molecules for 2–24 h and harvested at day 8. Fractionated primary adipocytes from iWAT and eWAT were obtained from varies group mice. Firstly, tissue was carefully dissected and minced into small pieces with scissors and digested with 0.05% collagenase type I (Roche) for 1 h at 37 °C with shaking at 100 rpm. Cell suspension was filtered through a 100 μM nylon filter to remove undigested tissue debris, followed by washing with 2 volumes of washing medium (DMEM/F12(1:1); Gibco) containing 10% FBS). The filtrates were then centrifuged at 500 $g$ for 6 min at room temperature, and the floating adipocytes were transferred and resuspended in washing medium before centrifugation for 3 times. The fractionated primary adipocytes were cultured overnight. Human primary adipocytes and SVF cells from individuals with obesity come from Renmin Hospital of Wuhan University (Wuhan, China) or are isolated from lipoaspirate waste donated post-surgery from women with obesity. Weigh the adipose tissue and rinse with 1x PBS pH 7.4. Cut 4 g of WAT and use scissors to mince the tissue into small pieces in a dissection tray. Transfer minced WAT to a sterile 50 mL centrifuge tube, add 20 mL of PBS, and shake gently to remove the excess of red blood cells. Transfer WAT to a new sterile 50 mL centrifuge tube and add 25 mL of digestion solution. Incubate at 37 °C for 60 min in an orbital shaker at 125 rpm. Filter the digested tissue through three layers of gauze into a new sterile 50 mL centrifuge tube. Centrifuge at 200 $g$ for 5 min at 4 °C. Cells were seeded, maintained until two days after becoming confluent, and differentiated into adipocytes in the presence of rosiglitazone and isobutylmethylxanthine.

## The sEV purification, characterisation and uptake

sEVs were purified by ultracentrifugation of conditioned medium[27]. Briefly, the conditioned medium was collected from cells grown in a medium containing sEVs-depleted serum for 48 h, and pre-cleared by centrifugation at 500 $g$ for 15 min and then at 12,500 $g$ for 20 min. sEVs were pelleted by ultracentrifugation at 110,000 $g$ for 70 min, and washed in PBS using the same ultracentrifugation conditions. The sEVs pellet was suspended in PBS and used in various experiments. Tissue-derived sEVs were separated following the protocol[54]. Specifically, a small (~50 mg) piece of tissue was weighed and briefly sliced on dry ice and then incubated in 100 U/mL collagenase type 1 in HANKS solution (Solarbio, H1025) at 37 °C. The dissociated tissue was spun at 300 $g$ for 10 min at 4 °C. Iodixanol/OptiPrep (Sigma-Aldrich, D1556) density gradient centrifugation was performed as described[27]. For cell treatment, 2 μg of sEVs (equivalent to those derived from ~5 × 10^6 parental cells) based on protein measurement using BCA protein assay kit (Thermo Fisher Scientific) was added to 2 × 10^5 recipient cells in one well from a six-well plate. To prepare mouse or human serum, whole blood was collected into separator tube (BD Biosciences). After blood coagulation, the tube was centrifuged at 1200 $g$ for 15 min to separate serum. For sEVs purification with human and mouse serum, 500 μL serum was mixed with 900 μL of PBS and centrifuged at 10,000 $g$ for 30 min; the supernatant was then mixed with 10 mL PBS before ultracentrifugation at 110,000 $g$ for 70 min to pellet sEVs.

EVs size distribution and particle concentration were analysed using Flow NanoAnalyzer U30 (NanoFCM, Xiamen, China) according to the manufacurer's instructions. Briefly, the instrument calibrated for particle concentration to 10^7–10^9 particles/mL with 250 nm fluorescent silica nanospheres (NanoFCM Inc., QS2503) and for size distribution using Silica Nanospheres Cocktail (NanoFCM Inc., S16M-Exo). The report for size distribution and particle concentration was generated directly from the NF Profession 2.0 software, with the particles from PBS subtracted. Flow cytometer (CytoFLEX, Beckman Coulter) verified the sEVs quantity of Rab27a knockout or knock down outcomes in 4T1 and MDA-MB-231 cells. The instrument calibrated for gating sEVs with 100 and 200 nm Latex Beads (PCS Control Mixed Kit, Beckman Coulter, 6602336). About 10^9 particles/mL fresh sEVs were diluted of 300-fold in PBS. The report for particle concentration was generated directly from the CytExpert 2.2 software.

The sEVs derived from breast cancer cells were labelled by the cell membrane labelling agent CFSE (eBioscience, 65-0850-84). After being seeded in 96-well plates and allowed to differentiate, mature 3T3-L1 cells were incubated with labelled exosomes (20 μL/well) for the indicated time. When indicated, CFSE was added into the PBS at 5 μM and incubated for 1 h at 37 °C before the washing step, followed by an extra round of washing in PBS to remove the excess dye. Images were acquired using the Olympus FluoView FV1000. To monitor the trafficking of sEVs containing miRNAs in vivo, Lck-GFP labelled sEVs were injected by tail vein, Lck-GFP were visualised with an Operetta High Content Imaging system (PerkinElmer, USA).

## RNA extraction and quantitative PCR with reverse transcription

Total RNA was harvested from cultured cells or tissues using the TRIzol reagent (Invitrogen, 15596026) followed by cDNA synthesis using the MonScript RTIII Super Mix with dsDNase kit (Monad Biotech, MR05201). A CFX Connect real-time PCR system (Bio-Rad Laboratories) was used to perform quantitative real-time PCR (RT-qPCR) of cDNA samples using MonAmp ChemoHS qPCR Mix (Monad Biotech, MQ00401). The exosomal RNAs were extracted by TRIzol LS reagent (Invitrogen, 10296028). sEVs resuspended in 1 mL TRIzol LS and then incubated at room temperature for 10 min. 200 μL of chloroform was added, gently mixed, and incubated for 10 min on ice. After incubation, microcentrifuge tubes were centrifuged for 10 min at 12,000 $g$ at 4 °C. The aqueous phase was transferred to new tubes, an equal volume of isopropanol and 1 μL glycogen (Roche, 10901393001) were added and mixed with incubation at −20 °C overnight. Samples were centrifuged at 12,000 $g$ for 15 min at 4 °C. Aspirated off liquid and added 500 μL of cold 75% ethanol to wash the pellet. The RNA pellet was air dried for a few minutes at room temperature and subsequently resuspended in 20 μL of RNase-free water. The cellular or exosomal miRNAs was reverse transcribed according to the miScript II RT Kit (QIAGEN, 218161) and measured using miScript SYBR Green PCR Kit (QIAGEN, 218073). The primers for miR-204-5p, miR-16-5p and U6 were purchased from Tiangen (Cat# CD201-0087, CD201-0235 and CD201-0145). All the other primers were designed using PrimerBank and the sequences are listed in Supplementary Table 2.

## Western Blot analysis

Cells and sEVs were harvested and extracted with NP-40 buffer (50 mM Tris-HCl, pH 7.4, 150 mM NaCl, EDTA, 1% Nonidet P-40, phosphatase inhibitor and protease inhibitor) for SDS-PAGE. Approximately 20 mg of tissues were ground with radio-immunoprecipitation assay buffer (50 mM Tris-HCl, pH 7.4, 150 mM NaCl, 1% Triton X-100, 1% sodium deoxycholate, 0.1% SDS, sodium orthovanadate, sodium fluoride, EDTA and leupeptin). For sEVs characterisation with immunoblotting, 200 ml cultured medium was used to extract sEVs and dissolved in 100 μL PBS. Total amount of 20 μg protein were loaded in each well. For protein assays on hypothalamic tissue, mice were euthanized by $CO_2$ and quickly decapitated, and the brains were removed. Using a brain block guide, the tissue rostral and caudal to the hypothalamus was dissected. Medial basal hypothalamus was then rapidly excised and placed in ice-cold homogenisation buffer. Proteins were separated by electrophoresis on a 10% or 12% SDS polyacrylamide gel. Protein levels were quantified with ImageJ software. Protein detection was performed using the antibodies described in Supplementary Table 3.

## Immunohistochemistry (IHC) and in situ hybridisation (ISH)

Adipose tissues were fixed in 10% neutral formalin for at least 48 h at 4 °C and then desiccated and embedded in paraffin before being cut into 5-μm sections before staining with H&E (Biosharp, BL700B). The *hsa*-miR-204-5p probe with 5'DIG and 3'DIG (5' AGGCAUAGGAUGA-CAAAGGGAA3') was purchased from Sangon Biotech (R11929). For IHC analysis, adipose tissues sections were mounted on adhesion slides, deparaffinized and rehydrated using a standard protocol. In situ hybridisation (ISH) were performed formaldehyde-fixed, paraffin-embedded tissue sections[55]. Stained slides were scanned using an Aperio VERSA digital pathology scanner (Leica Biosystems). The antibodies used were listed in Supplementary Table 3.

## Oil Red O staining

SVF-differentiated adipocytes were fixed with 4% paraformaldehyde (Sigma-Aldrich, P6148) in PBS and were washed twice with PBS. Completely air-dried at room temperature, cells were then incubated with Oil Red O working solution (Sigma, O0625) at room temperature for 1 h. Cells were washed 3 times with PBS before images were acquired for analysis. For quantification of Oil red O staining, cells were washed with PBS twice and then 200 μL DMSO was added to each well. After incubation for 10 min with gently shaking, samples were measured for OD at 510 nm.

Tissues were immediately frozen in liquid nitrogen and cut using a Leica CM-1850 cryostat microtome (Leica, Wetzlar, Germany). Afterwards, 16 mm-thick sections were fixed in 4% formaldehyde for 10 min and stained with filtered 0.5% Oil Red O, which was dissolved in isopropyl alcohol, for 15 min at room temperature.

## Measurements of fatty acid and TG content

Free fatty acid (FFA) levels in mice serum were measured using colorimetric assay kit (A042-2-1, Nanjing Jiancheng Bioengineering Institute, China) according to the instructions from the manufacturer. Triglyceride (TG) content was quantified using a colorimetric kit (Cayman, 10010303). The results were read at a wavelength of 570 nm by using an microplate reader (BioTek, Winooski, Vermont, USA).

## Determination of the lipid composition

Lipid composition was measured by MetWare (Wuhan, China) following the standard protocol. For lipidomics analysis, the sample was taken out from the −80 °C refrigerator, thawed on ice and vortexed for 10 s. Mix 50 μL of the sample and 1 mL of the extraction solvent (MTBE: MeOH =3:1, v/v) containing internal standard mixture. After whirling the mixture for 15 min, 200 μL of water was added. Vortex for 1 min and centrifuge at 16,000 g for 10 min. 200 μL of the upper organic layer was collected and evaporated using a vacuum concentrator. The dry extract was reconstituted using 200 μL mobile phase B prior to LC-MS/MS analysis. Then, lipids were solubilized in 2-propanol before UPLC analysis. Lipids were analysed on an Ultra High Performance Liquid Chromatography System (UPLC lc-30a) equipped with a Phenomenex Kinetex C18 column (100 × 2.1 mm, 2.6 μm) column. Lipids were eluted using a gradient from solvent A, $H_2O$: MeOH: CAN (1:1:1 containing 5 mM $NH_4Ac$), to solvent B, IPA: ACN (5:1 containing 5 mM $NH_4Ac$). The analysis was carried out at 60 °C with a flow rate of 0.4 mL/min. MS analysis was performed using the AB Sciex TripleTOF® 6600 System, operating in a positive ion mode and controlled by Analyst 1.6.3 software (AB Sciex). Glycolipids were quantified using PE as the standard. The KEGG enrichments for metabolites from serum and adipocytes were deposited in Figshare (https://figshare.com/s/c054d2c057f661b0b01a).

## Lentivirus transfection

HEK293T cells were transfected using Lipofectamine 2000 (Life Technologies) with packaging plasmid: psPAX2 (Addgene, #12260) and pMD2.G (Addgene, #12259). Forty-eight hours after transfection, lentiviral supernatant was harvested and filtered through a 0.45 μm filter, and utilised for subsequent infection of MCF-10A, MDA-MB-231 and 4T1, 3T3-L1 and SVF cells (48 hours incubation) together with 10 μg/mL polybrene (Beyotime, C0351) in the medium. Cells were selected based on puromycin selection or sorted by GFP+ cells with FACS.

## RNA-seq and Gene Set Enrichment Analysis (GSEA)

RNA sequencing was performed by Majorbio (Shanghai, China) and transcriptome library was prepared following TruSeq RNA sample preparation Kit from Illumina (San Diego, CA) using 1 μg of total RNA. Libraries were size selected for cDNA target fragments of 300 bp through 2% Low Range Ultra Agarose followed by PCR amplified using Phusion DNA polymerase (NEB) for 15 PCR cycles. After quantified by TBS380, paired-end RNA-seq sequencing library was sequenced with the Illumina HiSeq xten/NovaSeq 6000 sequencer (2 × 150 bp read length). The raw paired end reads were trimmed and quality controlled by SeqPrep (https://github.com/jstjohn/SeqPrep) and Sickle (https://github.com/najoshi/sickle) with default parameters. Then clean reads were separately aligned to reference genome with orientation mode using HISAT2 (https://daehwankimlab.github.io/hisat2) software. The mapped reads of each sample were assembled by StringTie (https://ccb.jhu.edu/software/stringtie/index.shtml?%20t=example) in a reference-based approach. The expression level of each transcript was calculated according to the transcripts per million reads (TPM) method. RSEM (http://deweylab.biostat.wisc.edu/rsem/) was used to quantify gene abundance. Eventually, differential expression analysis was performed using the DESeq2. Sequencing data are deposited in the NCBI Gene Expression Omnibus (GEO) under accession code GSE222380.

## Electron microscopy

Small EV pellets were resuspended in PBS filtered through a 0.22 μm filter, gently dropped on the copper grids for 1 min, and then stained using 2% uranyl acetate (SPI-Chem, #02624-AB) for 30 s at room temperature. Differentiated primary adipocytes were fixed in 2.5% glutaraldehyde in 0.1 M sodium cacodylate buffer pH 7.4 for 1 h prior to washing with 100 mM phosphate buffer without $CaCl_2$. Samples were then fixed in 1% osmium tetroxide for 1 h followed by staining with 2% uranyl acetate (SPI-Chem, #02624-AB) in maleate buffer pH 5.2 for 1 h. Samples were rinsed and dehydrated in a series of ethanol, and then embedded in resin (Embed 812 EMS) and baked overnight at 60°C. Hardened blocks were cut using a Leica Ultra Cut UCT, and 60 nm sections were collected on carbon-coated grids, followed by contrast-staining with 2% uranyl acetate and lead citrate. Embedded samples were analysed by JEM-1400 plus electron microscope at 80 KV (Joel Ltd, Tokyo, Japan).

## Respiration assays

Cellular oxygen consumption rate (OCR) and extracellular acidification rate (ECAR) of primary adipocytes was determined using an XFe96 Extracellular Flux Analyzer (Seahorse Bioscience). Primary adipocytes were seeded at $7 × 10^3$ per well cultured and differentiated in XF96 cell culture microplates (Agilent, 101085-004). 24 h later, adipocyte culture medium was changed to XF assay medium (Agilent, 103575-100) supplemented with 2 mM glutamine (for ECAR) (Agilent, 103579-100) or 2 mM glutamine, 10 mM glucose (Agilent, 103577-100), and 1 mM pyruvate (Agilent, 103578-100) (for OCR). Cells were incubated in a $CO_2$-free incubator for 1 h at 37 °C to allow for temperature and pH equilibration prior to loading into the XFe96 analyzer (Agilent, 103015-100) for measurement following the manufacturer's protocol. Basal glycolysis was calculated by subtracting the third baseline (non-glycolytic acidification) ECAR reading from the third glycolysis ECAR reading following glucose addition. Basal respiration was derived by subtracting the third OCR reading following antimycin A addition from

the third basal OCR reading. Uncoupled and maximal OCR was determined using Oligomycin and FCCP (4 μM each) (Agilent, 103015-100), respectively. Complex I-dependent respiration was inhibited with Rotenone (2 μM) (Agilent, 103015-100). A mitochondrial fuel flex test (Agilent, 103260-100) was performed to measure the dependency of adipocytes to oxidise three critical mitochondrial: glucose, glutamine, and FFAs. Adipocytes were exposed to BPTEs (3 μM), Etomoxir (4 μM), or UK5099 (2 μM) in succession, and OCR was measured prior to and after the addition of each compound. Data was assessed with XF Wave software 2.4. Protein was quantified as described, and all data were normalized to protein content.

### Transwell migration and invasion assay

Transwell chamber (8 μm pore size; Corning, USA) with or without matrigel (Corning, USA) were used to determine the invasion ability of BC cells. In brief, $2 \times 10^5$ cells were suspended in 500 μL DMEM containing 1% FBS and added to the upper chamber, while 750 μL DMEM containing 10% FBS was placed in the lower chamber. After 48 h of incubation, matrigel and the cells remaining in the upper chamber was removed using cotton swabs. Cells on the lower surface of the membrane were fixed in 4% paraformaldehyde and stained with 0.5% crystal violet (Sigma-Aldrich, 32675). Cells in 5 microscopic fields were counted and photographed. All experiments were performed in triplicate.

### CCK8 assay

Cell viability was analysed by Cell Counting Kit-8 (CCK8, Yeasen Biotechnology, 40203ES76) according to the manufacturer's protocols. Cells were seeded and cultured at a density of $5 \times 10^3$ cells/well in 100 μL of medium into 96-well microplates (Corning, USA). 10 μL of CCK-8 reagent was added to each well and then cultured for 2 h. All experiments were performed in triplicate. The absorbance was analysed at 450 nm using a microplate reader (BioTek, Winooski, Vermont, USA) using wells without cells as blanks. The proliferation of cells was expressed by the absorbance.

### Animals

All animal experiments were approved by the Institutional Animal Care and Use Committee at the College of Life Science, Wuhan University. Mice (*Mus musculus*) were maintained in 12 h light/dark cycles (6 am-6 pm) at 24 °C with 50–60% humidity and fed standard irradiated rodent chow diet. The maximal tumour size/burden is less than 1500 mm³ and a single tumour is less than 1.5 cm in any direction in our study and under the guidance of Institutional Animal Care and Use Committee at the College of Life Science, Wuhan University. Feed chow composition mainly includes corn grain,soybean meal, wheat flour, fish meal, soybean oil, multivitamin, multiple mineral elements and so on. Replenish the feed chow every 2-3 days. The 6–10 weeks old female mice were used for the animal experiments. Sample size, determined empirically via performing preliminary experiments, was chosen to be at least 4 to ensure that adequate statistical power will be achieved. Female NOD/SCID/IL2Rγ-null (NSG) mice (for sEV tail vein injection, MDA-MB-231 tumour models) were purchased from Shanghai Model Organisms Center, female and male BALB/c mice purchased from Center for Disease Control (CDC; Hubei, China) (for 4T1 sEV tail vein injection, 4T1 xenograft model and C26 tumour models), C57BL/6 mice (for sEV tail vein injection, LLC tumour models) were purchased from GemPharmatech (Nanjing, China). BKS-Leprem2Cd479/Gpt (BKS-db) mice (Strain NO.T002407) for sEV tail vein injection were purchased from GemPharmatech (Nanjing, China). The sEVs were injected into the tail vein semi-weekly for 5 weeks (~10 μg sEVs per injection per mouse). MDA-MB-231 xenograft models were established by injecting orthotopically $2 \times 10^5$ WT cells, 231/Rab27a KD cells with an equal volume of matrigel (Corning, 356234) into the fourth pair of mammary fat pad. 4T1 xenograft models were established by injecting

$2 \times 10^5$ WT cells, 4T1/Rab27a KO cells and 4T1/miR-204 KO cells with an equal volume of matrigel into the fourth pair of mammary fat pad. Six-week-old male C57BL/6 J mice were anaesthetised with isoflurane and injected with half million of LLC cancer cells per 100 μL phosphate buffered saline (PBS) subcutaneously into the right flank. Half million of C26 tumour cells per 100 μL PBS were subcutaneously injected into the right flank of isofluraneanesthetized 6 weeks old male BALB/c mice. Control mice were injected subcutaneously with 100 μL PBS into the right flank. Tumour volume was calculated using the formula (length × width²)/2. For VHL restoration in adipose tissue, AAV-VHL virus or Ctrl virus was injected into intraperitoneally and subcutaneous of both sides ($1.5 \times 10^{12}$ genomic copies per injection per site) 1 week after implantation of 4T1 cells. Mice were divided into treatment groups randomly while satisfying the criteria that average body weight in each group will be about the same. $2 \times 10^5$ of 4T1 cells per mouse were injected mammary fat pads over the flank. Mice received intraperitoneal injections of AAV-VHL and subcutaneous injections of the AAV-VHL. All mice were sacrificed in late light cycle (3–6 pm). Mice were housed individually in all tumour inoculation experiments and in groups in other experiments. To monitor tumour metastasis, mice which received intraperitoneal injection with D-luciferin (GoldBio) were subjected to bioluminescence imaging. Images were captured and the bioluminescence signal was quantified using IVIS Spectrum imaging system (PerkinElmer). Plasma was collected into EDTA tubes for the leptin ELISA assay (mlbio, ml002287). The adipose tissue NE levels were measured with NE ELISA assay (mlbio, ml063805). The hypothalamic tissues were quickly harvested from the mice and stored at −80 °C for further analysis of gene expression. Cold-challenge experiments were performed within climate-controlled cold rooms. Mice were singly placed in individual precooled cages without bedding at 4 °C. Mice had free access to precooled food and water. Rectal core body temperatures of mice were measured at ZT3 using a RET-3 rectal probe (Physitemp) attached to a BAT-12 Microprobe Thermometer (Physitemp). Whole-body energy metabolism was evaluated using a Comprehensive Lab Animal Monitoring System (CLAMS, Columbia Instruments). $CO_2$ and $O_2$ data were collected every 13 min for each mouse and were normalized to total body weight. Data on activity, heat generation and food intake were measured at more frequent intervals. For hematoxylin and eosin (HE) staining, adipose tissues were fixed in 4% formaldehyde, embedded in paraffin, and cut into 6 μm sections on slides. Mice were anesthetized with 2% isoflurane (RWD Life Science Co., Ltd., Shenzhen, China), and CT images were acquired by 3D micro-CT (Quantum GX micro-CT; PerkinElmer, America). The CT images were visualised and analysed using Analysis 14.0 Visualisation and Analysis software. Abdominal fat was measured from the base of the ensiform cartilage to the pelvic floor and was divided into iWAT and eWAT.

### Human serum specimens

Archived samples from patients with cancer patients and health controls used in this study were collected in accordance with Clinical Research Ethical Committee of Renmin Hospital of Wuhan University. All participants provided written informed consent. Archived fasting serum samples were collected from female patients with BC and female control subjects without cancer. 500 μL of serum sEVs were isolated using qEV original / 70 nm column (IZON, SP1). The qEV column was equilibrated by PBS solution filtered using a sterile 0.22 μm filter. Fresh serum samples were loaded onto the loading frit of the qEV column. The first 3 mL of default buffer was discarded and the following 1.5 mL buffer containing sEVs was collected. Patients with advanced breast cancers were included in the study because of their greater susceptibility to weight loss and cachexia associated with cancer. All participants were under the direct care of a medical oncologist and received standard chemotherapy for their disease and stage. None of the patients were receiving medications with the intent

of managing cachexia such as corticosteroids, progestational agents or cannabinoids. Participants were asked to fast for 12 h and to refrain from strenuous exercise and alcohol for 24 h prior to assessments, which included body composition, resting energy expenditure (REE) and a blood sample. Leptin ELISA assay (mlbio, ml028534) was used to determine blood leptin. Serum samples collected from normal subjects were used to set baseline for measuring blood leptin. Adipose tissue samples were taken from the abdominal subcutaneous and the intra-abdominal omental fat depots at defined locations during surgery. Adipose tissue was analysed (immediately frozen in liquid nitrogen after explantation) for histology and protein expression analyses. Clinical information was obtained from pathology reports, and the characteristics of the included cases are provided in Supplementary Table 4.

### Software
Equipment built-in software were used for data collection, including CFX Manager Software version 3.1 for RT-qPCR data, CytExpert 2.0 software was used for Flow cytometer and NF Profession 2.0 was used for nanoFCM. Wave Software 2.4 was used for Seahorse. Prism 9.4.1 was used for most data analysis and statistics. GSEA 4.2.0 was used for GSEA of RNA-seq data. ImageJ (1.52) was used to quantify western blot data. Aperio ImageScope (12.4.3.5008) was used to quantify HE, IHC and ISH data. GMS 3 was used for TEM data. HIASAT (v0.6.1), Bowtie (2.4.4), fastx toolkit (0.0.13) were used in RNA-seq data analysis. TargetScan 7.2 (https://www.targetscan.org/vert_72/) and miRWalk (http://mirwalk.umm.uni-heidelberg.de/) was used to predict miRNAs targeting VHL.

### Statistics and reproducibility
All quantitative data are presented as mean ± SEM unless otherwise stated. Unpaired two-tailed Student's t-tests were used for comparison of means of data between two groups. For multiple independent groups, one-way or two-way ANOVA with Dunnett multiple comparison test were used. Values of $P < 0.05$ were considered significant. Sample size was generally chosen based on preliminary data indicating the variance within each group and the differences between groups. All samples that have received the proper procedures with confidence were included for analyses. Animals were randomised before treatments. Western blots were repeated independently three times with similar results, and representative images were shown.

### Reporting summary
Further information on research design is available in the Nature Portfolio Reporting Summary linked to this article.

## Data availability
All data relating to this study can be found in the main text, figures, or supplementary information. Source data are provided with this paper. The RNA-seq data generated in this study are available in the NCBI Gene Expression Omnibus (GEO) (https://www.ncbi.nlm.nih.gov/geo/query/acc.cgi?acc=GSE222380). EV miRNA sequencing data was referred to previously reported data (GSE50429). The LC-MS/MS-detected metabolomic data had been deposited in Figshare (https://figshare.com/s/c054d2c057f661b0b01a). Source data are provided with this paper.

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

## Acknowledgements

This work was supported by the National Natural Science Foundation of China (82203590 for Y.W.) and the Ministry of Science and Technology of China (2021YFA0804803). We thank Dr. Jing Zhang from Frontier Science Center for Immunology and Metabolism of Wuhan University for kindly providing the expression plasmid of WT and dpA mutant of HIF1A, and pcDNA-3.1-FLAG-VHL. We thank Dr. Wei Song for providing LLC cells, Dr.Pengcheng Bu for providing C26 cells and Dr.Wenhua Li for providing MGC-803 and GSE-1 cells. We thank all the core facilities including microscopy core and electron microscopy facility from College of Life Science, ultracentrifugation platform provided by the State Key Laboratory of Virology from Wuhan University and the pathology core from Renmin hospital of Wuhan University.

## Author contributions

W.Y. conceived ideas. W.Y., Y.C.X. and Y.L. contributed to project planning. Y.H. and L.L. performed most of the experiments and analysed the data. Y.C., X.L., M.X.L., H.F.Z. and X.H.Z. assisted with data analysis, cell line construction and EV characterisation. S.Y.C. and S.H. assisted with tissue processing and histological analyses. S.Y.C. and J.J.L. assisted with clinical sample assembly and assessment. W.Y. and Y.H. wrote the manuscript.

## Competing interests

The authors declare no competing interests.
