## [Peer Review File · Nature Communications]

Cancer-cell-secreted miR-204-5p induces leptin signaling pathway in white adipose tissue to promote cancer-associated cachexiaREVIEWER COMMENTS

Reviewer #1 (Remarks to the Author):

The article illustrates that circulating miR-204-5p in small EVs induce hypoxia-mediated leptin signaling pathway to promote lipolysis and WAT browning. The authors also show KO of Rab 27a and miR-204-5p in murine and human breast cancer models reduced cancer-induced lipolysis. Additionally, the db/db mice model and human patients' studies further strengthened the role of miR-204-5p in promoting breast cancer-associated lipolysis. The findings are interesting and could have important clinical applications in the field of cancer cachexia. The reviewer appreciates the body of work done by the authors and fortunately, all the assays seem to have worked. Some concerns remain and need to be addressed prior to publication.

Major concerns:

1. Cancer cachexia also affects muscle as illustrated by the authors. Though I appreciate the article is focused on adipose tissue, the effect of miR-204-5p containing small EVs on cancer-associated muscle wasting needs to be discussed as it is a key feature of cancer-associated cachexia. When the whole body weight is looked at in several figures in the manuscript, the contribution of muscle vs adipose tissue vs others is not clear and hence examination of muscle atrophy by miR-204 is critical.
2. Cachexia in breast cancer is not as common as in other types of cancer. Hence, it is important to examine the levels of miR-204-5p in high cachexia prevalent cancer models such as gastric and/or pancreatic cancers. Ideally, the examination of cell lines would suffice as the authors have already done a great job with breast cancer.
3. Several studies reported that miR-204-5p as a tumor suppressor in breast (PMID: 30737233) and a variety of human cancers and extended fig 5b showed that KO of miR-204 in 4T1 breast cancer cells significantly reduced tumor volume compared to the 4T1Ctrl? Does not this contradict the published literature?
4. Is the observed reduced lipolysis in 4T1/miR-204 KO group due to the smaller tumor volumes? It has been known in the cachexia field that tumor volumes positively correlate with weight loss. Is there data with the same amount of tumor volume but differential lipolysis? Please include tumor volumes with 4T1 and MDA-MB-231 in vivo studies.
5. Fig 1c-d, 6e-f: Please include the cachectic status and/or tumor volumes in the oxygen consumption and heat production fig description.
6. EV concentration, dose, and protein amount for western all can be included in methods and in figure legends for clarity.
7. 4T1 and MDA-MB-231 are metastatic breast cancer models. Did authors observe any benefit in metastatic burden upon loss of miR-204-5p? If yes, how does this affect the observed preservation of lipolysis and browning compared to their respective controls?
8. From Fig 7b, can levels of miR-204 in EVs correlate with the aggressiveness of cachexia? If not, how will the authors correlate with patient data?

Minor comments:

1. Please follow the MISEV 2018 guidelines (PMID: 30637094), while characterising EVs.
2. Please correct line 89 to fluorescence (fig 1f).
3. Please include details of miR-204 and control mimics.
4. Line 247-249: EV educated adipocytes data is missing in extended fig 5d-e. Please include.
5. Most of the full names are not mentioned. Please include the full names whenever they are used for the first time (example Mt-Co1/NDUFV1).
6. Line 316: please correct "acturally" to actually.
7. Fig 7i – cachexia spelling mistake.

Reviewer #2 (Remarks to the Author):

The manuscript titled "Cancer-cell-secreted miR-204-5p induces leptin signaling pathway in white adipose tissue to promote cancer associated cachexia" presents interesting results on the

mechanism of the induction of adipose tissue wasting in response to breast cancer tumors. The major findings include:

- a. 4T1 breast cancer xenografts elevated energy expenditure and induced white adipose tissue (WAT) wasting while Rab27-knockout 4T1 cells defective for EV secretion induced less dramatic effects. These effects were accompanied by the transfer of a membrane-localized GFP protein from the 4T1 cells to adipose tissues, which was reduced for Rab27-KO cells. EVs isolated from the MDA-MB-231 breast cancer cells but not from the MCF10A non-cancer cells induced similar effects, arguing a role for EVs in adipose tissue wasting.
- b. MDA-MB-231 tumors and their EV secretion suppressed VHL expression and induced HIF1 α protein levels in white adipose tissues. miR-204 present in the EV secretion was shown to suppress VHL expression. In fact, miR-204 overexpression in MCF10A xenografts induced adipose tissue wasting while its depletion in 4T1 cells suppressed hypermetabolism and wasting of adipose depots.
- c. Leptin expression in WAT and leptin plasma levels were induced by 4T1 xenografts and MDA-MB-231-derived EVs, which led to a response in the hypothalamus, including the upregulation of STAT3 phosphorylation.
- d. 4T1 xenografts and MDA-MB-231-derived EVs expressing miR-204 induced thermogenesis, WAT browning and lipolysis, accompanied by increased mitochondrial content and activity. Leptin receptor-deficient mice were resistant to the metabolic changes.
- e. Restoration of VHL expression in WAT of 4T1 tumor-bearing mice suppressed HIF1 α , UCP1 and LEPTIN proteins and the thermogenic activity of WAT.
- f. Elevated Leptin levels were detected in plasma of breast cancer patients and elevated miR-204 was reported in adipose tissue of cachectic breast cancer patients, which also had higher UCP1 protein and HSL phosphorylation, indicative of enhanced thermogenesis and lipolysis, respectively.

These findings argue that a tumor signaling to white adipose tissue mediated by miR-204 carrying EVs leads to a hypoxic response in WAT and activates leptin signaling promoting thermogenesis, lipolysis and wasting of WAT depots. While the results presented in the manuscript are very convincing, authors should consider the following comments to improve their work.

Major comments:

1. Authors performed experiments using 4T1 xenografts and MDA-MB-231-derived EVs. It is important that the same experiments are repeated for 4T1-derived EVs and MDA-MB-231 xenografts and the effects on energy metabolism and adipose tissue thermogenesis and lipolysis are characterized. MDA-MB-231 xenografts were used in Fig. 2 but changes in tissue weight were not reported.
2. In all tumor inoculation experiments, authors should report changes in weight of BAT, skeletal muscle tissue (GA/TA) and tumors. e.g. weight of all tissues is missing in Fig 2.
3. Changes in BAT weight and gene expression is missing in tumor inoculation and EV treatment experiments in Fig. 2, 3, 4, 5 and 6. Particularly, in Fig. 4, a dramatic effect on body temperature was reported. Since BAT is the major fat depot carrying out nonshivering thermogenesis, authors should report their observations on this tissue. Even if BAT is not picking up tumor-derived EVs from the circulation to the same extent as WAT depots, it is important to see how its behavior changes in the experiments concerning energy metabolism.
4. In Extended Data Fig 2o, authors showed that food intake was lower in mice receiving MCF-10/miR-204 EV and it was higher in mice with 4T1/miR-204-KO xenografts. It is not clear if miR-204 has an impact on food consumption and if this effect is linked to the Leptin signaling. Authors should provide food intake information for all metabolic cage experiments, including Fig. 1c-d and 1j-d and Fig. 6e-f. Changes in body weight should also be reported. If food intake is significantly

reduced in 4T1 xenograft-bearing mice or MDA-MB-231-EVs-receiving mice, authors should comment on the relative impact of changes in energy metabolism and food intake on the loss of WAT mass.

5. In Extended Data Fig. 4, wild-type controls of db/db mice is missing. In the methods part, it was stated that NOD/SCID/IL2RKO mice were used for the EV injection experiments. Mice from this background is not a good control for db/db mice. Authors should perform EV injection experiments in C57BL/6 mice to demonstrate that the reported phenotypes also manifest in immunocompetent mice.

6. In Extended Data Fig. 5b, authors showed that 4T1 tumors were larger than tumors lacking Rab27A and miR-204. It is possible that differences in tumor growth confound the data on energy metabolism and adipose wasting presented in Fig 1, 3 and 6. It is important that authors provide tumor weight data in these experiments. If tumor weight is different between experimental groups, authors should comment on the potential confounding effects. In Extended Data Fig. 6b-c, a big difference in tumor volume was reported, which did not reflect on the tumor mass. This is not understood.

7. To show that the findings presented in the manuscript are not limited to breast cancer and immunodeficient mice, authors should consider using a syngeneic graft model of cachexia, such as LLC lung, B16 melanoma and C26 colon, and perform tumor inoculation experiments in immunocompetent mice.

8. In Fig. 3, GSEA showing the enrichment of Leptin pathway elements is not convincing and should be removed (FDR values are too high). A very small gene set was used here. Authors may try to expand their set.

9. In Fig. 5f-g, the use of H&E staining of WAT depots is not sufficient to claim enhanced browning. Authors should perform UCP1 IHC as in Fig. 6m-n.

10. In Extended Data Fig. 4, authors described the lack of miR-204-mediated WAT browning in leptin receptor-deficient mice. UCP1 blots should be added to 4i. Authors should also emphasize that although miR-204 containing EVs do induce HIF1a and LEPTIN expression and suppress VHL expression in WAT of db/db mice, the metabolic effects were missing in these mice lacking the Leptin signaling (Page 9).

Minor comments:

11. I believe the language of the manuscript can be improved. Grammar and punctuation errors should be corrected.

12. Statistical analysis was not described in the legend of Fig. 7.

13. Color designation of the micro-CT visuals is missing in Fig. 1h and Extended Data Fig. 4j.

14. It will be very helpful if the text lines are numbered throughout the manuscript.

15. In page 2, the word "cytoxin" should be double-checked. Should it be "cytokine"?

16. In Fig. 2i, miR-204 mimic was used. This molecule should be described at least in the methods section. It is important to know if this corresponds to human or mouse miR-204.

17. In page 8, authors stated that "Inversely, when we overexpressed VHL to suppress the expression of HIF1A in 3T3-L1 cells, the abundance of HIF1A was sharply reduced and lost the responsiveness by miR-204 EV (Fig. 3i)". However, the cells were still responsive to miR-204 containing EVs as HIF1a and Leptin levels increased. This argument should be clarified.

18. In page 10, is it AAV8 or AAV9?

19. In page 10, authors stated that "GFP signals were detected in adipose tissue rather than other organs or xenograft tumor of AAV-GFP-injected mice (Extended Data Fig.6a)". This is confusing because GFP signal was present in GA and liver and the tumor signal was not shown.

20. Extended Data Fig. 2 title is misleading. miR-204 is not inducing hypoxia but it appears to promote hypoxia signaling/hypoxia response.

21. In Extended Data Fig. 2f and 6a, muscles abbreviated as GA and TA must be described.

22. In page 11, authors stated that "TB AAV-VHL mice exhibited more food intake than TB AAV-GFP mice and displayed higher body temperature (Figs.6c and 6d)". This should be corrected as lower body temperature. It is also stated that "Besides, energy expenditure was also reversed under the overexpression of VHL (Figs.6e and 6f), behaving with similar RER and locomotor activity (Extended Data Figs.6e and 6f)". However, RER appears to be higher and locomotor activity is lower in the TB AAV-GFP group. Please clarify.

23. In page 12, authors stated that "As circulating miR-204 has been associated with weight loss in non-cancer patient". References are missing here.

24. In Fig. 7c-e, authors utilized human primary adipocytes and SVF cells. Extraction and culturing procedures should be described in the methods section.

25. In Fig. 7j-k, authors compared "control adipose tissue" with "tumour associated adipose tissue". It is not clear if these samples come from the same patients and they have different proximity to the tumor. Were these samples collected from non-cancer patients vs breast cancer patients? Clarification is needed here.

Response to Reviewers

Reviewer 1

The article illustrates that circulating miR-204-5p in small EVs induce hypoxia-mediated leptin signaling pathway to promote lipolysis and WAT browning. The authors also show KO of Rab 27a and miR-204-5p in murine and human breast cancer models reduced cancer-induced lipolysis. Additionally, the db/db mice model and human patients' studies further strengthened the role of miR-204-5p in promoting breast cancer-associated lipolysis. The findings are interesting and could have important clinical applications in the field of cancer cachexia. The reviewer appreciates the body of work done by the authors and fortunately, all the assays seem to have worked. Some concerns remain and need to be addressed prior to publication.

Response: Thank you very much for reviewing our paper and really appreciate your constructive comments during your busy schedule. The comments and suggestions are all valuable and very helpful. We believe that the quality of our manuscript has greatly improved with your help. Following your valuable suggestion, we amended the relevant text in the manuscript and please see the point-to-point responses shown below.

Major comments:

1. Cancer cachexia also affects muscle as illustrated by the authors. Though I appreciate the article is focused on adipose tissue, the effect of miR-204-5p containing small EVs on cancer-associated muscle wasting needs to be discussed as it is a key feature of cancer-associated cachexia. When the whole body weight is looked at in several figures in the manuscript, the contribution of muscle vs adipose tissue vs others is not clear and hence examination of muscle atrophy by miR-204 is critical.

Response: Thank you very much for your valuable advices. Following your suggestion, we have discussed the effect of miR-204-5p on skeletal muscle loss in the discussion section of our revised manuscript (lines 369-382, page 15). We added: "As cancer-associated cachexia is characterized as a multi-organ wasting disorder with mass loss in both adipose tissue and skeletal muscle, we also observed the reduced skeletal muscle mass under miR-204-5p EV treatment but this effect was fully blocked in db/db mice (Extended Data Figure 4b), suggesting the regulation of circulating miR-204 on skeletal muscle atrophy highly depends on the lipolysis of adipose tissue. Of note, the reduced muscle mass percentage is far less than that in adipose tissue. Specifically, we observed a ~1.0% decrease for iWAT and eWAT while a 0.15% reduction for GA and TA muscle in 4T1 or MDA-MB-231 xenografted tumor mice. Parallely, we found more fat loss than skeletal muscle mass loss in mice receiving MDA-MB-231 EV or 4T1 EV (Extended Data Figures 1f and 1k). Therefore, the effect of BC-derived miR-204 on cachexia results more from the lipolysis of adipose tissue rather than muscle atrophy. Although the direct effect and mechanism of miR-204-5p on skeletal muscle have not been examined in our study, it has been reported that miR-204-5p inhibits myoblast differentiation by targeting myocyte enhancer factor 2C (MEF2C) and estrogen-related receptor gamma (ERR γ) (PMID: 29505923)".

Extended Data Figure 1f

Extended Data Figure 1k

2. Cachexia in breast cancer is not as common as in other types of cancer. Hence, it is important to examine the levels of miR-204-5p in high cachexia prevalent cancer models such as gastric and /or pancreatic cancers. Ideally, the examination of cell lines would suffice as the authors have already done a great job with breast cancer.

Response: Thanks a lot for your advice. Following your suggestion, we have examined some high cachexia prevalent cancer cell lines including MGC-803 (gastric cancer), LLC (lung cancer) and C26 (colon cancer). Indeed, we detected higher levels of miR-204-5p in intracellular and EV fractions when compared to the respective control group (Figures 8c-e). To further investigate the expression of miR-204-5p under different cancer types in vivo, we constructed both LLC and C26 tumor models, and found there were 9.12- and 3.23-fold higher in tumor-bearing mice serum, along with more fat loss (Figure 7 and Extended Data Figure 7).

3. Several studies reported that miR-204-5p as a tumor suppressor in breast (PMID: 30737233) and a variety of human cancers and extended fig 5b showed that KO of miR-204 in 4T1 breast cancer cells significantly reduced tumor volume compared to the 4T1Ctrl? Does not this contradict the published literature?

Response: Thanks for pointing out this important question. We seriously considered the

question raised by Reviewer 1 and thoroughly read the conflicting data from past literature (PMID: 30737233). Actually, the introduction section of their study mentions that the role of miR-204-5p in breast cancer is controversial (“While some studies showed upregulation of miR-204-5p in breast cancer tissues and the pro-proliferative role of miR-204-5p in breast cancer cells in vitro”, page 1520 *Cancer Res*; 79(7)). In their study, three of four tumors with miR-204 overexpression displayed no significant alteration of tumor growth rate and tumor weight when compared to the control group (Figures 3b-c). According to their description, even one cell line with the highest miR-204 overexpression level (C#4: 1419.0 ± 37.7) displayed a faster tumor growth rate than another cell line (C#3: 525.0 ± 62.1), which was exactly in line with our observation (page 1524 and Figure 3a). Moreover, additional studies also supported the upregulation of miR-204-5p in breast cancer tissues and the pro-proliferative role of miR-204-5p in breast cancer cells (PMID: 18922924; PMID: 27121770). Therefore, miR-204-5p might play a complicated role in tumor growth under different stages. Although we did observe miR-204-5p KO tumor has a slower tumor growth rate, our study still focuses on the pathophysiologic significance of circulating miR-204-5p level derived from cancer cells.

4. Is the observed reduced lipolysis in 4T1/miR-204 KO group due to the smaller tumor volumes? It has been known in the cachexia field that tumor volumes positively correlate with weight loss. Is there data with the same amount of tumor volume but differential lipolysis? Please include tumor volumes with 4T1 and MDA-MB-231 in vivo studies.

Response: Really appreciate your informative comments. We did find a positive correlation between tumor volume and weight loss especially in 4T1/miR-204 KO and MDA-MB-231/miR-204 KO group mice when compared to their respective control groups (Extended Data Figures 5b and 8f). However, in another C26 xenograft tumor model, we found C26/miR-204 KO group mice with similar tumor volume but endowed with differential lipolysis when compared to wild type tumor-bearing mice (Figure 7m), suggesting the weight loss was tightly related to the level of miR-204-5p rather than tumor volume. We also discussed this in lines 382-387, page 15.

5. Fig 1c-d, 6e-f: Please include the cachectic status and/or tumor volumes in the oxygen consumption and heat production fig description.

Response: Thank you for your comments. As you suggested, we have added the change of body weight data which indicates cachectic status (Extended Data Figure 1c, Figure 6a) and tumor volume data (Extended Data Figures 5b and 6b). We also added the related description when we described oxygen consumption and heat production (line 81, page 4

and line 287, page 11).

6. EV concentration, dose, and protein amount for western all can be included in methods and in figure legends for clarity.

Response: Thanks for your helpful suggestions. We have included those details in the method section and figure legends, highlighted in the revised manuscript (line 520, page 20; line 575, page 21; line 1424, page 63). For sEVs characterization with immunoblotting, 200 mL cultured medium was used to extract sEVs and dissolved in 100 μ L PBS. A total amount of 20 μ g protein was loaded in each well. For cell treatment, 2 μ g of sEVs (equivalent to those derived from $\sim 5 \times 10^6$ producer cells) based on protein measurement using a BCA protein assay kit was added to 2×10^5 recipient cells in a six-well plate. For mice treatment, sEVs were injected into the tail vein semi-weekly for five weeks (~ 10 μ g sEVs per injection per mouse).

7. 4T1 and MDA-MB-231 are metastatic breast cancer models. Did authors observe any benefit in metastatic burden upon loss of miR-204-5p? If yes, how does this affect the observed preservation of lipolysis and browning compared to their respective controls?

Response: Thanks for your constructive question. We performed a live animal imaging assay to monitor the metastasis (indicated by luciferase activity) of 4T1/miR-204 KO and 231/miR-204 KO cells, and observed significantly reduced metastatic capability compared to their respective control groups (Extended Data Figures 8d-e). Our current data is not sufficient to support the underlying connection between lipolysis and tumor metastasis. However, some studies have reported that adipokines, fatty acids and other metabolites from lipolysis may promote tumor metastasis (PMID:22037646, PMID:35541892). There is a high possibility that decreased metastasis in 4T1/miR-204 KO and 231/miR-204 KO mice may result from reduced lipolysis.

Extended Data Figures 8d and 8e

8. From Fig 7b, can levels of miR-204 in EVs correlate with the aggressiveness of cachexia? If not, how will the authors correlate with patient data?

Response: Thanks for your question. Although we examined the miR-204 level in EVs from distinct breast cancer (BC) cell lines and confirmed higher miR-204 amount is a common phenomenon in breast cancer-derived EVs, we could not conclude that miR-204

in BC EVs correlates with the aggressiveness of cachexia due to lack of reported data linking the occurrence of cachexia between distinct breast cancer types. However, this suggested higher circulating miR-204-5p levels might universally distribute among distinct BC types (lines 338-339, page 13). To link the clinical data, we examined the levels of miR-204-5p from BC patient serum and compared the expression with non-BC patients and patients with cancer-associated cachexia (Figures 8i-l).

Minor comments:

1. Please follow the MISEV 2018 guidelines (PMID: 30637094), while characterizing EVs.

Response: Thanks for your advice. Referred to the MISEV 2018 guidelines (PMID: 30637094), we have introduced detailed information for treating cells and animals. For cell treatment, 2 µg of sEVs (equivalent to those derived from ~5×10⁶ producer cells) based on protein measurement by BCA protein assay kit, was added to 2×10⁵ recipient cells seeded in one well from a six-well plate. For in vivo EV treatment, sEVs were injected into the tail vein semi-weekly for five weeks (~10 µg sEVs per injection per mouse). For sEVs characterization with immunoblotting, we loaded 20 µg of sEVs and examined three positive markers and one negative marker. We also examined the size distribution by Flow NanoAnalyzer (line 530, page 20 and Extended Data Figures 8a-c). Besides, we have added sEVs imaged by electron microscopy in Extended Data Figures 8a-c.

2. Please correct line 89 to fluorescence (fig 1f).

Response: Thanks for pointing out this error. We have corrected to fluorescence in the revised manuscript (line 95).

3. Please include details of miR-204 and control mimics.

Response: Thanks for your suggestion. We have added detailed information of miR-204 and control mimics (line 476).

4. Line 247-249: EV educated adipocytes data is missing in extended fig 5d-e. Please include.

Response: We really appreciate for your suggestions. We have added EV educated adipocytes data in the Extended Data Figures 5g-h.

5. Most of the full names are not mentioned. Please include the full names whenever they are used for the first time (example Mt-Co1/NDUFV1).

Response: Thanks for this valuable comment. We have carefully proofread the entire manuscript and added corresponding full name and replaced the abbreviations were

introduced for the first time.

6. Line 316: please correct “acturally” to actually.

Response: Thanks for pointing out this spelling mistake. We have corrected “acturally” to actually in line 388.

7. Fig 7i – cachexia spelling mistake.

Response: Thanks for pointing out this spelling. We have corrected “Figure 8i–cachexia” on line 1210.

Reviewer 2

Remarks to the Author

The manuscript titled “Cancer-cell-secreted miR-204-5p induces leptin signaling pathway in white adipose tissue to promote cancer associated cachexia” presents interesting results on the mechanism of the induction of adipose tissue wasting in response to breast cancer tumors. The major findings include:

- a. 4T1 breast cancer xenografts elevated energy expenditure and induced white adipose tissue (WAT) wasting while Rab27-knockout 4T1 cells defective for EV secretion induced less dramatic effects. These effects were accompanied by the transfer of a membrane-localized GFP protein from the 4T1 cells to adipose tissues, which was reduced for Rab27-KO cells. EVs isolated from the MDA-MB-231 breast cancer cells but not from the MCF10A non-cancer cells induced similar effects, arguing a role for EVs in adipose tissue wasting.
- b. MDA-MB-231 tumors and their EV secretion suppressed VHL expression and induced HIF1a protein levels in white adipose tissues. miR-204 present in the EV secretion was shown to suppress VHL expression. In fact, miR-204 overexpression in MCF10A xenografts induced adipose tissue wasting while its depletion in 4T1 cells suppressed hypermetabolism and wasting of adipose depots.
- c. Leptin expression in WAT and leptin plasma levels were induced by 4T1 xenografts and MDA-MB-231-derived EVs, which led to a response in the hypothalamus, including the upregulation of STAT3 phosphorylation.
- d. 4T1 xenografts and MDA-MB-231-derived EVs expressing miR-204 induced thermogenesis, WAT browning and lipolysis, accompanied by increased mitochondrial content and activity. Leptin receptor-deficient mice were resistant to the metabolic changes.
- e. Restoration of VHL expression in WAT of 4T1 tumor-bearing mice suppressed HIF1a, UCP1 and LEPTIN proteins and the thermogenic activity of WAT.
- f. Elevated Leptin levels were detected in plasma of breast cancer patients and elevated miR-204 was reported in adipose tissue of cachectic breast cancer patients, which also had higher UCP1 protein and HSL phosphorylation, indicative of enhanced thermogenesis and lipolysis, respectively.

These findings argue that a tumor signaling to white adipose tissue mediated by miR-204 carrying EVs leads to a hypoxic response in WAT and activates leptin signaling promoting thermogenesis, lipolysis and wasting of WAT depots. While the results presented in the manuscript are very convincing, authors should consider the following comments to improve their work.

Response: Thanks very much for reviewing our work and for your constructive comments

during your busy schedule. The comments and suggestions are much valuable and very helpful for improving and polishing our paper. We have considered your precious comments carefully and have made corresponding corrections and hopefully, the revision will answer your questions. Overall, the point-to-point responses are seen below.

1. Authors performed experiments using 4T1 xenografts and MDA-MB-231-derived EVs. It is important that the same experiments are repeated for 4T1-derived EVs and MDA-MB-231 xenografts and the effects on energy metabolism and adipose tissue thermogenesis and lipolysis are characterized. MDA-MB-231 xenografts were used in Fig. 2 but changes in tissue weight were not reported.

Response: Thanks for your comments. Following your suggestions, we performed the parallel experiments in mice receiving 4T1-derived EVs and MDA-MB-231 xenografts mice and summarized the data in Figure 1 and Extended Data Figure 1. Mice receiving 4T1-derived EVs and MDA-MB-231 xenografts mice exhibited significantly stronger lipolysis, adipose tissue thermogenesis and energy metabolism when compared to their respective control groups. We apologize for missing the MDA-MB-231 xenografts tissue weight changes in Figure 2. In our revised manuscript, we updated Extended Data Figure 1f with the addition of the tissue weight data.

2. In all tumor inoculation experiments, authors should report changes in weight of BAT, skeletal muscle tissue (GA/TA) and tumors. e.g. weight of all tissues is missing in Fig 2.

Response: Thanks again for your valuable comments. We have supplemented the results including the weight of BAT, skeletal muscle tissue (GA/TA) and tumors of MDA-MB-231 and 4T1 xenografted mice and updated in Extended Data Figures 1f, 5c, 6c, 7c and 7n and Figures 6b,7c and 7n.

3. Changes in BAT weight and gene expression is missing in tumor inoculation and EV treatment experiments in Fig. 2, 3, 4, 5 and 6. Particularly, in Fig. 4, a dramatic effect on body temperature was reported. Since BAT is the major fat depot carrying out nonshivering thermogenesis, authors should report their observations on this tissue. Even if BAT is not picking up tumor-derived EVs from the circulation to the same extent as WAT depots, it is important to see how its behavior changes in the experiments concerning energy metabolism.

Response: We really appreciate your helpful comments and totally agree that BAT-related data should also be included in the manuscript for a better understanding of the systemic effect of EVs on whole-body metabolism. We have included BAT weight and gene expression from tumor inoculation and EV treatment experiments (Extended Data Figures 1k and 7j, Figures 4d, 6g and 7j). The main thermogenic gene containing *UCP1*, *Pgc1a*, *Prdm16* and *Dio2* have an elevation in 10A/miR-204 EV group mice and Tumor-bearing group mice.

4. In Extended Data Fig 2o, authors showed that food intake was lower in mice receiving MCF-10/miR-204 EV and it was higher in mice with 4T1/miR-204-KO xenografts. It is not clear if miR-204 has an impact on food consumption and if this effect is linked to the Leptin signaling. Authors should provide food intake information for all metabolic cage experiments, including Fig. 1c-d and 1j-d and Fig. 6e-f. Changes in body weight should also be reported. If food intake is significantly reduced in 4T1 xenograft-bearing mice or MDA-MB-231-EVs-receiving mice, authors should comment on the relative impact of changes in energy metabolism and food intake on the loss of WAT mass.

Response: Thanks again for those useful suggestions. We have provided food intake data for all metabolic cage experiments and body weight in Extended Data Figures 1c,1d,1i and 1l and discussed in page 16 of the main text. We detected food intake was significantly reduced in 4T1 xenograft-bearing mice or MDA-MB-231-EVs-receiving mice, which may be partially due to the elevated leptin signaling as it can directly inhibit appetite by regulating the activity of AgRP and POMC in the hypothalamic neurons of the brain (PMID: 17937601). However, miR-204 EV injection failed to regulate food intake in db/db mice, which may suggest the effect of circulating miR-204 on food intake is leptin-dependent.

5. In Extended Data Fig. 4, wild-type controls of db/db mice is missing. In the methods part, it was stated that NOD/SCID/IL2RKO mice were used for the EV injection experiments. Mice from this background is not a good control for db/db mice. Authors should perform EV injection experiments in C57BL/6 mice to demonstrate that the reported phenotypes also manifest in immunocompetent mice.

Response: Thanks again for your suggestion and we totally agree that C57BL/6 mice rather than NOD/SCID/IL2RKO mice serve as better control for db/db mice. Following your suggestion, we have performed 10A/miR-204 EV injection experiments in C57BL/6 mice and updated the data in Extended Data Figure 4. The results indicate that 10A/miR-204 EV can promote fat lipolysis, adipose tissue thermogenesis and energy metabolism in immunocompetent mice, which is blocked in db/db mice due to the lack of leptin receptors.

6. In Extended Data Fig. 5b, authors showed that 4T1 tumors were larger than tumors lacking Rab27A and miR-204. It is possible that differences in tumor growth confound the data on energy metabolism and adipose wasting presented in Fig 1, 3 and 6. It is important that authors provide tumor weight data in these experiments. If tumor weight is different between experimental groups, authors should comment on the potential confounding effects. In Extended Data Fig. 6b-c, a big difference in tumor volume was reported, which did not reflect on the tumor mass. This is not understood.

Response: Thanks for raising those critical questions. We have discussed the effect of tumour weight on energy metabolism and adipose wasting in the discussion section of the revised manuscript (lines 384-387, page 15). We have added all mice tumor weight Figure 7n and Extended Data Figures 5c, 6c, 7n and 8f. We have observed smaller tumor volumes and tumor weights in 4T1/miR-204 KO and MDA-MB-231/miR-204 KO mice when compared to their respective control groups. In our breast cancer model, a smaller tumor weight may affect fat lipolysis with reduced miR-204 secretion, as our miR-204 EV-educated mice displayed an enhanced lipolysis rate. For extended Data Figures 6b-c, we have added extra four mice into each group and reanalyzed all tumor weight data in the revised manuscript (Extended Data Figure 6c). The results showed that the tumor weight

of 4T1 was significantly greater than those tumors lacking the expression of Rab27A and miR-204.

7. To show that the findings presented in the manuscript are not limited to breast cancer and immunodeficient mice, authors should consider using a syngeneic graft model of cachexia, such as LLC lung, B16 melanoma and C26 colon, and perform tumor inoculation experiments in immunocompetent mice.

Response: Thank you very much for your valuable advice. Following your suggestion, we have constructed both lung (LLC) and colon (C26) cancer models by xenografted tumor-bearing and miR-204 KO cells to confirm the generalizability of the mechanism besides breast cancer (Figure 7 and Extended Data Figure 7). Compared with the wild-type mice, miR-204 KO mice in both lung and colon cancer models showed decreased lipolysis and energy metabolism, which extends to breast cancer.

Figure 7

Extended Data Figure 7

8. In Fig. 3, GSEA showing the enrichment of Leptin pathway elements is not convincing and should be removed (FDR values are too high). A very small gene set was used here. Authors may try to expand their set.

Response: Thank you very much for your valuable advice. We have expanded the gene sets by merging original Reactome signaling by leptin (11 genes) and the WP leptin signaling pathway (25 genes) and shown in Figure 3a.

9. In Fig. 5f-g, the use of H&E staining of WAT depots is not sufficient to claim enhanced browning. Authors should perform UCP1 IHC as in Fig. 6m-n.

Response: Thanks for your useful comment. Following your suggestion, we have added UCP1 IHC in iWAT/eWAT from each group of mice (Figures 5h and 5i).

Figure 5h

Figure 5i

10. In Extended Data Fig. 4, authors described the lack of miR-204-mediated WAT browning in leptin receptor-deficient mice. UCP1 blots should be added to 4i. Authors should also emphasize that although miR-204 containing EVs do induce HIF1a and LEPTIN expression and suppress VHL expression in WAT of db/db mice, the metabolic effects were missing in these mice lacking the Leptin signaling (Page 9).

Response: Thanks for your valuable comment. We have added UCP1 blots in Figure 4i. Besides, we have revised this page and emphasized that miR-204 containing sEVs promoted HIF1A and LEPTIN expression in WAT of db/db mice, but the metabolic effects were missing in these mice lacking the leptin signaling (line 247, page 10).

Extended Data Figure 4i

Minor comments:

11. I believe the language of the manuscript can be improved. Grammar and punctuation errors should be corrected.

Response: Thanks for your valuable comments. We apologize for the language issue. We have carefully improved the language and corrected all grammar and punctuation errors in the manuscript.

12. Statistical analysis was not described in the legend of Fig. 7.

Response: Thanks for this advice. We have added statistical analysis in the legend of Figure 8.

13. Color designation of the micro-CT visuals is missing in Fig. 1h and Extended Data Fig. 4j.

Response: Thanks for this valuable comment. We have added color designation of the micro-CT visuals in Figure 1j and Extended Data Figure 4k.

14. It will be very helpful if the text lines are numbered throughout the manuscript.

Response: Thanks for your advice. We agreed with your suggestion and added text lines numbered throughout the manuscript.

15. In page 2, the word “cytoxin” should be double-checked. Should it be “cytokine”?

Response: Thanks for your advice. We are sorry about this mistake. We have corrected “cytoxin” to “cytokine” in page 2 of the manuscript.

16. In Fig. 2i, miR-204 mimic was used. This molecule should be described at least in the methods section. It is important to know if this corresponds to human or mouse miR-204.

Response: We are sorry about this confusion. We have added miR-204 mimic molecule in the method section (line 476, page 18).

17. In page 8, authors stated that “Inversely, when we overexpressed VHL to suppress the expression of HIF1A in 3T3-L1 cells, the abundance of HIF1A was sharply reduced and lost the responsiveness by miR-204 EV (Fig. 3i)”. However, the cells were still responsive to miR-204 containing EVs as HIF1a and Leptin levels increased. This argument should be clarified.

Response: We are sorry about this confusion. We have reconstructed overexpressing VHL in 3T3-L1 cells and repeatedly confirmed the expression of the HIF1a and Leptin (Figure 3i). Protein quantification showed no significant increase in the miR-204 EVs group compared to the PBS group.

18. In page 10, is it AAV8 or AAV9?

Response: Thanks for your advice. We are sorry about this mistake. We have corrected “AAV8” to “AAV9” in page 11 of the revised manuscript.

19. In page 10, authors stated that “GFP signals were detected in adipose tissue rather than other organs or xenograft tumor of AAV-GFP-injected mice (Extended Data Fig.6a)”. This is confusing because GFP signal was present in GA and liver and the tumor signal was not shown.

Response: Thanks for providing this important comment. We apologize for not describing Extended Data Figure 6a clearly. We have revised the description in page 11 and added the data related to tumour signal in Extended Data Figure 6a.

20. Extended Data Fig. 2 title is misleading. miR-204 is not inducing hypoxia but it appears to promote hypoxia signaling/hypoxia response.

Response: Thanks for your correction. We have corrected the title of the Extended Data Figure 2.

21. In Extended Data Fig. 2f and 6a, muscles abbreviated as GA and TA must be described.

Response: Thanks for your advice. We have carefully proofread the entire manuscript and replaced the abbreviations mentioned initially with full name. All changes in the manuscript text file are highlighted in blue.

22. In page 11, authors stated that “TB AAV-VHL mice exhibited more food intake than TB AAV-GFP mice and displayed higher body temperature (Figs.6c and 6d)”. This should be corrected as lower body temperature. It is also stated that “Besides, energy expenditure was also reversed under the overexpression of VHL (Figs.6e and 6f), behaving with similar

RER and locomotor activity (Extended Data Figs.6e and 6f)". However, RER appears to be higher and locomotor activity is lower in the TB AAV-GFP group. Please clarify.

Response: Thank you for correcting us. We have changed the description (line 286, page 11) (Extended Data Figures 6c and 6d) and fixed this mistake (line 289, page 12) in the Extended Data Figures 6e and 6f.

23. In page 12, authors stated that "As circulating miR-204 has been associated with weight loss in non-cancer patient". References are missing here.

Response: Thanks for raising this important comment. We are sorry about this mistake. We have removed this sentence in the Revised manuscript. There have been no previous literature reports on miR-204 being associated with weight loss, only with muscle atrophy.

24. In Fig. 7c-e, authors utilized human primary adipocytes and SVF cells. Extraction and culturing procedures should be described in the methods section.

Response: Thanks for pointing out this. We have added human primary adipocyte extraction and culturing procedures in the methods section (line 504, page 19). We added the description below: Human primary adipocytes and SVF cells from overweight and obese female donors come from Renmin Hospital of Wuhan University (Wuhan, China) or are isolated from lipoaspirate waste donated post-surgery from women with obesity using methods as previously described (PMID: 18370084). Cells were seeded, maintained until two days after becoming confluent, and differentiated into adipocytes in the presence of rosiglitazone and isobutylmethylxanthine as previously described.

25. In Fig. 7j-k, authors compared "control adipose tissue" with "tumour associated adipose tissue". It is not clear if these samples come from the same patients and they have different proximity to the tumor. Were these samples collected from non-cancer patients vs breast cancer patients? Clarification is needed here.

Response: We are sorry about this confusion. In our study, "control adipose tissue" came from non-cancer patients and "tumour-associated adipose tissue" were collected from breast cancer patients. We have updated detailed information in the legend of Figures 8j-k and main text (line 1211, page 48).

REVIEWERS' COMMENTS

Reviewer #1 (Remarks to the Author):

The authors have addressed all the concerns raised in the initial review.

Reviewer #2 (Remarks to the Author):

I thank the authors for revising their manuscript in line with the comments provided and also for the detailed explanation of the changes made. All of my concerns were addressed in the revised manuscript. However, I have the following minor comments:

1. In line 226, authors state that "Moreover, primary adipocytes incubated with 10A/miR-204 sEVs or MDA-MB-231 sEVs displayed a higher oxygen consumption rate but lower glycolytic rate than PBS or MCF-10A sEVs treated adipocytes (Figs.4i and 4j)". However, the figure legend indicates "(i) Oxygen consumption rate (OCR) and (j) extracellular acidification rate (ECAR) in primary white adipocytes isolated from iWAT of each group mice". It should be clarified if the adipocytes were treated with the EVs or if they were extracted from EV-treated mice.

2. Similarly, In line 258, authors suggest "Moreover, electron microscopy analysis showed tumour-derived miR-204 sEVs induced reduction in lipid droplet size in differentiated primary adipocytes". This is again confusing. It is not clear whether differentiated adipocytes were treated with EVs or these adipocytes came from EV-treated mice.

In line 200, authors argued that incubation with EVs fail to induce lipolysis in cultured adipocytes because leptin mediated crosstalk between brain and adipose tissue is needed. Therefore, EV treatment of cultured adipose cell should not alter oxygen consumption or lipid droplet size.

Response to Reviewers

Reviewer 1

The authors have addressed all the concerns raised in the initial review.

Response: Thank you very much for reviewing our paper and really appreciate your recognition of our revision.

Reviewer 2

I thank the authors for revising their manuscript in line with the comments provided and also for the detailed explanation of the changes made. All of my concerns were addressed in the revised manuscript. However, I have the following minor comments:

1. In line 226, authors state that “Moreover, primary adipocytes incubated with 10A/miR-204 sEVs or MDA-MB-231 sEVs displayed a higher oxygen consumption rate but lower glycolytic rate than PBS or MCF-10A sEVs treated adipocytes (Figs.4i and 4j)”. However, the figure legend indicates “(i) Oxygen consumption rate (OCR) and (j) extracellular acidification rate (ECAR) in primary white adipocytes isolated from iWAT of each group mice”. It should be clarified if the adipocytes were treated with the EVs or if they were extracted from EV-treated mice.

Response: Thanks very much for your constructive comments and your efforts during your busy schedule. The comments are much valuable and very helpful for improving the quality of our paper. We are sorry about this confusion due to the unclear description in the main text. These primary adipocytes were derived from EV-treated mice rather than in vitro EV treatment. We have revised the description and updated the detailed information in the revised manuscript (line 245, page 10).

2. Similarly, In line 258, authors suggest “Moreover, electron microscopy analysis showed tumour-derived miR-204 sEVs induced reduction in lipid droplet size in differentiated primary adipocytes”. This is again confusing. It is not clear whether differentiated adipocytes were treated with EVs or these adipocytes came from EV-treated mice.

In line 200, authors argued that incubation with EVs fail to induce lipolysis in cultured adipocytes because leptin mediated crosstalk between brain and adipose tissue is needed. Therefore, EV treatment of cultured adipose cell should not alter oxygen consumption or lipid droplet size.

Response: Thanks for your questions and comments. We totally agree that EV treatment of cultured adipose cell should not alter oxygen consumption or lipid droplet size. Indeed, these primary adipocytes came from EV-treated mice rather than in vitro EV treatment. Again, we are so sorry for this confusion. We revised the related description and updated the detailed information in the revised manuscript (line 280, page 11).